



# A segmentation algorithm for characterizing Rise and Fall segments in seasonal cycles: an application to XCO$_2$ to estimate benchmarks and assess model bias

Leonardo Calle[1], Benjamin Poulter[2], and Prabir K. Patra[3]

[1] Montana State University, Department of Ecology, Bozeman, Montana 59717, USA

[2] NASA Goddard Space Flight Center, Biospheric Science Laboratory, Greenbelt, Maryland 20771, USA

[3] Japan Agency for Marine-Earth Science and Technology (JAMSTEC), Yokohama, 236-0001, Japan

*Correspondence to*: Leonardo Calle (leonardo.calle@montana.edu)



**Abstract.** There is more useful information in the time series of satellite-derived column-averaged carbon dioxide ($XCO_2$) than is typically characterized. Often, the entire time series is treated at once without considering detailed

features at shorter timescales, such as non-stationary changes in signal characteristics - amplitude, period, and phase. In many instances, signals are visually and analytically differentiable from other portions in a time series. Each *Rise* (increasing) and *Fall* (decreasing) *segment*, in the seasonal cycle is visually discernable in a graph of the time series. The Rise and Fall segments largely result from seasonal differences in terrestrial ecosystem production, which means that the segment's signal characteristics can be used to establish observational benchmarks because the signal

characteristics are driven by similar underlying processes. We developed an analytical segmentation algorithm to characterize the Rise and Fall segments in $XCO_2$ seasonal cycles. We present the algorithm for general application of the segmentation analysis and emphasize here that the segmentation analysis is more generally applicable to cyclic time series.

We demonstrate the utility of the algorithm with specific results related to the comparison between

satellite- and model-derived $XCO_2$ seasonal cycles (2009-2012) for large bioregions on the globe. We found a seasonal amplitude gradient of 0.74-0.77 ppm for every 10˚ degrees of latitude for the satellite data, with similar gradients for Rise and Fall segments. This translates to a south-north seasonal amplitude gradient of 8 ppm for $XCO_2$, about half the gradient in seasonal amplitude based on surface site in-situ $CO_2$ data (~19 ppm). The latitudinal gradients in period of the satellite-derived seasonal cycles were of opposing sign and magnitude (-9

days/10˚ latitude for Fall segments, and 10 days/10˚ latitude for Rise segments), and suggests that a specific latitude (~ 2˚ N) exists which defines an inversion point for the period asymmetry. Before (after) the point of asymmetry inversion, the periods of Rise segments are less (greater) than the periods of Fall segments; only a single model could reproduce this emergent pattern. The asymmetry in amplitude and period between Rise and Fall segments introduces a novel pattern in seasonal cycle analyses, but while we show these emergent patterns exist in the data,

we are still breaking ground in applying the information for science applications. Maybe the most useful application is that the segmentation analysis allowed us to decompose the model biases into their correlated parts of biases in amplitude, period, and phase, independently for Rise and Fall segments. We offer an extended discussion on how such information on model biases and the emergent patterns in satellite-derived seasonal cycles can be used to guide future inquiry and model development.

**KEYWORDS:** GOSAT, DGVM, segmentTS, time series analysis, land use change, seasonal cycle



## 1. Introduction

Most of our understanding about atmospheric $CO_2$ dynamics has come from $CO_2$ sampled by in-situ flask samples or eddy-flux towers at Earth's surface (Ciais et al., 2014). While these data streams have proved incredibly useful, the transient dynamics of fluxes simulated by global-scale terrestrial models have only been compared to a relatively

few locations on Earth. In contrast to surface $CO_2$ samples, which sample $CO_2$ concentrations in the planetary boundary layer, satellite observations of $CO_2$ are made by downward-looking Fourier spectrometers from the top of the atmosphere and represent an integrated estimate of $CO_2$ concentrations in a full column of atmosphere, hereafter 'XCO$_2$' (Wunch et al., 2011; Crisp et al., 2012). Although fluxes from the surface have a large influence on the total column $CO_2$, the vertical and horizontal transport of air masses in higher atmospheric layers, each with different

concentrations $CO_2$, also influences the $CO_2$ concentrations in the total column (Belikov et al., 2017), including that of the stratosphere (Saito et al., 2012).

The synoptic coverage and integrated nature of XCO$_2$ means that surface fluxes from around the globe impart information into the seasonal dynamics and inter-annual variability of regional seasonal cycles, which is both a confounding and useful property for evaluating large-scale models. The integrated nature of the data also means that

even a few years of data will be sufficient to evaluate the simulated dynamics of global-scale models. We propose that if models can reasonably simulate the timing and magnitude of terrestrial surface fluxes in all bioregions, then we would expect that the simulated XCO$_2$ would match reasonably well with the seasonal dynamics from the benchmark satellite data. Such demonstrated ability could strengthen confidence in regional-to-global model simulations.

To gain insight into seasonal cycle dynamics of satellite XCO$_2$ and individual model behavior, we demonstrate a novel approach to extract more information from the seasonal cycle than is typically characterized. In evaluations of model performance, traditional performance statistics (root-mean-squared-error, correlation, standard deviation) are used to quantify bias in phase and amplitude of the seasonal cycle against a benchmark signal (Coupled Model Intercomparison Project (CMIP) Earth System Models *in* Glecker et al., 2008; DGVMs *in* Anav et al., 2015). In

almost all applications, however, the entire time series is treated at once without considering detailed features at shorter timescales, such as non-stationary changes in amplitude, magnitude, period, or phase (Fig. 1). We suggest that traditional performance statistics be applied to categories of unique patterns in the seasonal cycle, and not to the entire time series, thereby characterizing the error structure in a manner that can relate temporal dynamics (amplitude, magnitude, phase) with unique underlying processes.

We extend and apply a time series segmentation method (Ehret and Zehe, 2011) to extract the Rise and Fall segments in seasonal cycles of satellite-derived and simulated XCO$_2$, based on a suite of terrestrial ecosystem models. The advantage of the segmentation approach is that it allows an error structure to be accurately characterized by separately calculating the errors in amplitude, period and phase for each segment type (Rise, Fall). For example, in a graph of a multi-year seasonal cycle of XCO$_2$ (Fig. 1), each *increasing* and *decreasing* segment is

visually discernable and analytically differentiable from other portions in the seasonal cycle; hereafter, *Rise* refers to increasing segments and *Fall* refers to decreasing segments in a seasonal cycle. The Rise and Fall segments largely result from seasonal differences in the onset and cessation of terrestrial ecosystem production (Keeling et al., 1995),



which means that a segment's signal characteristics (i.e., amplitude, period, phase) are likewise influenced by different stages of terrestrial ecosystem activity. By segmenting the time series into similar component signals, we

can then test for differences in the signal characteristics of Rise and Fall patterns and provide insight into a model's ability to recreate these features of the seasonal cycle over multiple years.

Our first aim was to simply characterize the satellite-derived $XCO_2$ seasonal cycles in terms of Rise- and Fall-type segment variation. Secondly, we evaluated if signal characteristics and model biases differed or were correlated among Rise and Fall segments, which would help provide information in the missing parts of the satellite-based

time-series (i.e., at high latitudes during boreal winter and in the Tropics during the wet-season), which we demonstrate is possible. We also evaluated if model biases between Rise and Fall segments differed enough to provide information about the underlying model representation of terrestrial dynamics, which we underscore as possible but discuss the limits for inference in this regard. Lastly, we explored how a single modeled process (land use and land cover change; LUC) manifests in the different signal characteristics and biases in Rise and Fall

segments. We offer discussion on how the segment-based model biases and emergent patterns in satellite-derived seasonal cycles can be used to guide future inquiry and model development.

## 2. Methods

### 2.1 Satellite $XCO_2$ data

Satellite observations of $XCO_2$ were obtained from the Greenhouse gases Observing SATellite (GOSAT; version

7.3). Onboard the satellite, a Fourier-transform spectrometer measures the thermal and near-infrared absorption spectra of the constituent atmospheric gases within the footprint of observation (~10 km). Data for 2009-2012, corresponding to the timeframe of available simulation data, were freely obtained from NASA Goddard Earth Sciences (GES) Data and Information Services Center (DISC) online repository (<https://oco2.gesdisc.eosdis.nasa.gov/data/GOSAT_TANSO_Level2/ACOS_L2_Lite_FP.7.3/>; accessed 25 April

2018). We used the Level-2 *Lite* data products, which include only high-quality and bias-adjusted data points, based on the Atmospheric $CO_2$ Observations from Space (ACOS) retrieval algorithm version 7.3 (Crisp et al., 2012; O'Dell et al., 2012).

### 2.2 Simulated Terrestrial Fluxes from DGVMs

The Net Biome Exchange (NBP) from land-to-atmosphere was simulated by six terrestrial ecosystem models (Table

1) that were part of the TRENDY model inter-comparison project version 2 (Sitch et al., 2015; dgvm.ceh.ac.uk). We use the atmospheric convention and make fluxes to the atmosphere positive, and fluxes to the land negative. We assumed that the primary modes of seasonal variability in terrestrial NBP at large scales is described by three terms, Net Ecosystem Production (Net Primary Production – Heterotrophic Respiration), fluxes from fire, and land use change (LUC). The modeled NBP underwent forward-transport model simulation using the atmospheric general

circulation model (AGCM) to obtain a simulated estimate of $XCO_2$, matching the spatial and time-frame of the observations (i.e., 'co-location' sampling). The protocol for the DGVM inter-comparison standardized the (i) forcing data: gridded (0.5°) climate (air temperature, short- and long-wave radiation, cloud cover, relative humidity and



precipitation), global annual mean $CO_2$; and the (ii) initial conditions for time-varying simulations for the past

century (1860-2012). We used simulated NBP for two sets of model simulations, one where land use (natural vegetation, crop, and pasture fractional cover) is fixed at values from the year 1860 ('S2' scenario described *in* Sitch et al., 2015), and another simulation where land use change is simulated according to the HistorY Database of the global Environment (HYDE v3.1; Goldewijk et al., 2011) ('S3' scenario as described *in* Sitch et al., 2015); both simulation types were forced with time-varying climate and $CO_2$.

### 2.3 Fossil Fuel and Ocean Fluxes

The modes of variability (trend, seasonality, intra- and inter-annual variability) in $XCO_2$ are also influenced by fluxes from oceanic exchange, fossil fuel consumption and cement production. We used a simplified model of oceanic $CO_2$ exchanges from Takahashi et al. (2009), and monthly-mean fossil fuel emissions from the European Commission's Emissions Database for Global Atmospheric Research (EDGAR v. 4.2), based on country-level reporting and emissions factors, and the Fossil Fuel Data Assimilation System (http://edgar.jrc.ec.europa.eu/).

**2.4 Simulated $XCO_2$ using an Atmospheric Model**

Simulations of atmospheric $CO_2$ were conducted for the period of 2009-2012 using the land, ocean, and fossil fuel fluxes. We used the Center for Climate Systems Research/National Institute for Environmental Studies/Frontier Research Center for Global Change (CCSR/NIES/FRCGC) AGCM-based chemistry transport model (ACTM) (Patra et al., 2009). The ACTM was run at a horizontal resolution of T106 (~1.125° X 1.125°), and 32 sigma-

pressure vertical levels. The simulated $XCO_2$ values were obtained by taking the sum of the pressure-weighted $CO_2$ concentrations over all vertical layers, equivalent to the column-averaged observations. We then sampled the ACTM $XCO_2$ data to match the spatial location, time (13 hr, local time) and date of the satellite observations. We obtained the simulated $XCO_2$ for each component flux (land, fossil fuel, ocean) and finally summed the components to get the $XCO_2$ used in bias evaluations.

**2.5 Extraction of $XCO_2$ Seasonal Cycles**

We first estimated the mean of daily $XCO_2$ values by averaging gridded values within each of the 11 TransCom region (Fig. 2), for both the observed and modeled $XCO_2$. This procedure was as straight forward as written above, and the accompanying computer code (software: R for Statistical Computing) is provided as additional Supplementary Material. We then applied a digital filtering algorithm (*ccgcrv* by Thoning et al., 1989;

https://www.esrl.noaa.gov/gmd/ccgg/mbl/crvfit/crvfit.html) to the mean time series to extract the long-term trend and seasonal cycles, fitted as a 2-term polynomial (linear growth rate was used because the time series spanned only 3 years) and a 4-term harmonic function to account to seasonal asymmetry. Temporal data gaps were linearly interpolated by the algorithm. After subtracting the long-term trend and seasonal cycle, the *ccgcrv* algorithm filters the residuals in the frequency domain using a Fast-Fourier Transform (FFT) algorithm to retain short- and long-term

interannual variation (additional details in Nakazawa et al., 1997; Pickers and Manning 2015). The cutoff for the short-term filter was set at the recommended value of 80 days (Thoning et al., 1989). The cutoff for the long-term



filter was set to a large number (3000), which is longer than the number of days in our time series (365 days/yr * 3 yr= 1095 days) because, with such a short time series, we needed to force the estimation of a linear trend with no interannual variation; otherwise, the algorithm would be too sensitive and derive variation in the trend without

practical justification. For all analyses here forth, we combined the seasonal cycle with the digitally filtered short-term variation and used the derived data points along the smoothed seasonal cycle curves for analysis.

### 2.6 Technical description of algorithm: Segmentation of seasonal cycles

The purpose of this section is to describe the technical algorithms used in the analysis. These algorithms are based on concepts put forth by Ehret and Zehe (2011), translated herein to the R computing language (R Development

Core Team, 2008). Where Ehret and Zehe (2011) focused on the single hydrological events, we modify and restructure the algorithm to accommodate much longer non-stationary cyclic time series for general application to seasonal cycle analyses. An R package for the segmentation algorithm is freely available at the GitHub repository <https://github.com/lcalle/segmentTS>.

### 2.6.1 Categorize segments and isolate seasonal Rise and Fall cycles

We first determine the first derivatives numerically. The *ccgcrv* signal decomposition algorithm outputs a daily time series in the form of a multi-dimensional array, but we focus on a subset of the array, the 2-dimensional rectangular matrix representing points along the detrended seasonal cycle,

$$\boldsymbol{B} = \begin{bmatrix} b_{1,1} & b_{1,2} & b_{1,3} \\ \vdots & \vdots & \vdots \\ b_{n,1} & b_{n,2} & b_{n,3} \end{bmatrix} \tag{1}$$

, where the first column is the row index, the second column are dates, the third column is the detrended $XCO_2$ ppm

with the short-term variation added back-in, the rows are the triplets of the index, time in the x-direction, and magnitude ($XCO_2$ ppm) in the y-direction.

       We can numerically determine the first derivative in the y-direction at each point via differencing, as in,

$$\nabla b_{i,2} = b_{i,2} - b_{i-1,2} \tag{2}$$

We then classifying each row in first column ($b_{i,2} \dots b_{n,2}$) into one of the following categories below and expand **B**

to a *n* X 4 matrix to store the classified values. The main objective is to classify the endpoints (Trough, Peak) of the Rise and Fall segments:

$$\forall i \in \{1 \dots n\}, \ \mathbf{b_{i,4}} = \begin{cases} Trough, & (\nabla b_{i,2} < 0) \wedge (\nabla b_{i+1,2} > 0) \\ Rise, & (\nabla b_{i,2} < 0) \wedge (\nabla b_{i+1,2} < 0) \\ Fall, & (\nabla b_{i,2} > 0) \wedge (\nabla b_{i+1,2} > 0) \\ Peak, & (\nabla b_{i,2} > 0) \wedge (\nabla b_{i+1,2} < 0) \\ Null, & otherwise \end{cases} \tag{3}$$

We then take the subset of endpoints (*S*) in the classified matrix **B**,

$$S \subset \mathbf{B} = \{\mathbf{B} \mid \mathbf{b}_{i,3}: Trough, Peak \tag{4}$$

, where *S* retains the dimensions of the **B**. A unique segment (*s*) is defined as a set of two consecutive endpoints (rows) in *S* that alternate in their classification of Trough or Peak, meeting the condition:





$$s \subset S = \{S | (S_{i,4}: Trough \; \wedge \; S_{i+1,4}: Peak) \; \vee \; (S_i: Peak \; \wedge \; S_{i+1,4}: Trough) \tag{5}$$

We identify local minima and maxima that are deviations in otherwise longer (seasonal) and more general Rise and Fall patterns based on two criteria below, and then reclassify the segments based on the class of the

segment with the largest amplitude. The amplitude of a segment ($a_s$) is defined as:

$$a_s = |s_{1,2} - s_{2,2}| \tag{6}$$

, where $s_{1,2}$ is the first endpoint in the second column (XCO$_2$ ppm), either a Trough or a Peak, and $s_{2,2}$ is the second endpoint for the specific segment, which, by definition the first endpoint must be classified ($s_{1,4}$) as one of Peak or Trough and must not have the same classification as the second endpoint ($s_{2,4}$).

The first criterion sets a minimum threshold for the amplitudes, redefining the set of endpoints defining the segments, as below:

$$s^* \subset s = \{s \; | \; a_s > minimum \; threshold \tag{7}$$

Segments that represent local minima or maxima that are not of interest to the user can be identified by a comparison of amplitudes of consecutive segments, dropping the segment with the lowest amplitude, as below:

$$s^{*\prime} \subset s^* = \{s^* \; | \; s^* \neq min(a_{s-1}, a_s, a_{s+1}) \tag{8}$$

This procedure results in a new subset of segment endpoints ($s^{*\prime}$) with consecutive elements that have similar classification (e.g., $s^{*\prime}_{1,4} := Peak$, and also, $s^{*\prime}_{2,4} := Peak$), which needs to be rectified. We keep the endpoints with the lowest *Trough* value and the largest *Peak* value,

$$s[t]^* \subset \begin{Bmatrix} s[t]_{1,2} & s[t+1]_{1,2} \\ s[t]_{2,2} & s[t+1]_{2,2} \end{Bmatrix} =$$

$$\begin{Bmatrix} s[t]^*_{1,2} = \begin{cases} min(s[t]_{1,2}, s[t+1]_{1,2}), & s[t]_{1,4} := Trough \wedge s[t+1]_{1,4} := Trough \\ max(s[t]_{1,2}, s[t+1]_{1,2}), & s[t]_{1,4} := Peak \quad \wedge s[t+1]_{1,4} := Peak \end{cases} \\ s[t]^*_{2,2} = \begin{cases} min(s[t]_{2,2}, s[t+1]_{2,2}), & s[t]_{2,4} := Trough \wedge s[t+1]_{2,4} := Trough \\ max(s[t]_{2,2}, s[t+1]_{2,2}), & s[t]_{2,4} := Peak \quad \wedge s[t+1]_{2,4} := Peak \end{cases} \end{Bmatrix} \tag{9}$$


, where $s[t]$ is a unique segment in the set of $s$ segments, $s[t+1]$ is the following consecutive segment, $s[.]_{1,2}$ and $s[.]_{2,2}$ are the segment first and last endpoints, respectively, and $s[t]^*$ is the updated segment with new endpoints $s[t]^*_{1,2}$ and $s[t]^*_{2,2}$, while segments $s[t], s[t+1]$ have been removed from the set of segments ($s$).

A (subjective) limit can also be set to exclude or include segments based on temporal proximity. For example,

consecutive minima (*minima*/maxima/*minima*) should not be considered local minima if separated by 365 days; these are probably real local minima driven by processes unique to different seasons. By contrast, local minima separated by 60 days may represent signals within the overall seasonal Rise and Fall pattern (e.g., due to fire). For this study, we are more interested in assessing the general seasonal patterns. We therefore estimate the temporal distance, in 'days' ($D_s$), between the first endpoints of consecutive segments and evaluate the condition as below,

$$D_s = s[t+1]_{1,3} - s[t]_{1,3}, given \; s[t]_{1,4} \; \wedge s[t]_{1,4} \; are \; of \; the \; same \; class \; (Trough, Peak) \tag{10}$$

$$s^* \subset s = \{s \; | \; D_s > minimum \; threshold \tag{11}$$

, where $s[.]_{1,3}$ is the endpoint date in the x-direction, and the minimum threshold for distance between endpoints is set at a conservative 250 days (~8 months), ensuring that only the main Rise and Fall patterns within a given year are captured. This conditional evaluation also results in a new subset of segments ($s^*$) with consecutive elements



with similar classification, as above, but Eq. 9 can be re-applied to select the endpoints which represent general Rise and Fall patterns.

Additional criteria can be applied to automate the removal of local minima/maxima that are not relevant to the user, but we caution that visual inspection of the signal is important to avoid unwanted reclassification of segments in the time series.

### 215 2.6.2 Human-assisted pattern recognition via visual inspection

The procedure outline in Sect. 2.6.1, above, is applied to both the reference ($R$) and modeled ($M$) seasonal cycle time series. In the best of cases, the procedure would result in matrices for $R$ and $M$, each with an equal number segments and the same sequence of endpoint classes (*Trough, Peak, Trough, Peak, ...*). In practice, however, the number and sequence of segments in $M$ will not always equal the number or sequence of segments in $R$. When variability in the 220 modeled seasonal cycle results in many local minima/maxima, and therefore many short Rise/Fall segments, there can be a mismatch between the indices of segments, wherein smaller/shorter segments in $M$ are matched to much larger/longer segments in $R$; this is simply an artefact from automation of the procedure outlined previously. Although we have implemented automated procedures in the algorithm that reconcile these types of mismatches, we found that it was considerably quicker to (*i*) conduct a 'blind' run of the algorithm on the data, (*ii*) visually inspect 225 the automated graphical plots of the seasonal cycles for mis-matching segments (Supplementary Material Fig. S1), (*iii*) identify the index of the mis-matching endpoints in $M$, and then finally (*iv*) re-run the algorithm specifying the index of the endpoint in $M$ for removal.

### 2.6.3 Segment signal characteristics and error statistics

The amplitude (Eq. 6) and period ($p$ in 'days') for all segments are first characterized, with the period defined as,

$$p_s = s_n - s_0 \tag{12}$$

, where $s_n$ and $s_0$ are the end and start dates of a segment, respectively. Then, for each segment in $M$ and $R$, a complementary vector $Mx$ and $Rx$ is created in the x-direction with a fixed number of, and equally-spaced, dates,

$$x = (x_1 \ldots x_k) \tag{13}$$

Each element in $Mx$ corresponds, by index, to an element in $Rx$, such that a matching pair exists. Similarly, a 235 complementary vector $My$ and $Ry$ is created in the y-directions, with the length of the vector matching the length of the vector in the x-directions ($k$). For each element in $My$ and $Ry$, we perform a linear interpolation of the values of $XCO_2$ ppm in **B** (**b**.,2) for the indices given by the dates in $Mx$ and $Rx$; fortunately, the linear interpolation is automated by the *approx* function in R, which makes this procedural step straightforward. The end result is, for every segment in $M$ and $R$, four vectors of equal length in $Mx, My$ and $Rx, Ry$, with the timing of the data and 240 values of $XCO_2$ ppm that follow the corresponding seasonal cycles in **B**. We can then decompose the corresponding errors in phase and magnitude along the time series,

$$Timing\ error = Mx - Rx \tag{14}$$

$$Magnitude\ error = My - Ry \tag{15}$$


Although in this paper we focus only on errors in amplitude, period, and phasing of the segments, the time series of errors in timing and magnitude are an additional level of detail in the error structure that is evaluated by the segmentation algorithm.

### 2.7 Statistical summaries

For each of the Rise and Fall segments within a region, we summarized the characteristics by averaging the amplitude (ppm), period (days), and the phase, which we estimated in two ways based on the day of year for the first

and last endpoint of the corresponding segment (DOYstart, DOYend, respectively). For model biases, we used the total sum of the component tracers (land + fossil fuel + ocean) and we summarized model biases as the region-average of segment-to-segment differences between model and observation. Although we aggregate the biases among segment types, and therefore lose information, we do this to demonstrate that there are distinct general patterns in the Rise and Fall segments, regardless of region. Of course, one might be more interested in one

bioregion over another, and while this is indeed possible and suggested, such analysis is not the intent of this paper.

The latitudinal variation of amplitude and period length for Rise and Fall segments was evaluated by comparing the regionally-averaged metrics against the average latitude of each TransCom region. We sought to evaluate a model's ability to reproduce the north to south gradient in seasonal cycle characteristics. We also use data from in-situ [CO2] flask samples for 2005-2015 (NOAA/ESRL/GMD CCGG cooperative air sampling network;

https://www.esrl.noaa.gov/gmd/ccgg/flask.php) as a check to evaluate latitudinal variations of surface site seasonal amplitudes. Surface sites were selected if they had a minimum of five years of data between 2005-2015, with at least one flask sample per month. The peak-trough amplitude was then taken from monthly averaged data. Linear correlations were deemed statistically significant at levels of $p=0.05$.

The amplitude and period length asymmetries between Rise and Fall segments were calculated as in the

following example. Given a sequence of data with segments of type {*Fall_1*, *Rise_1*, *Fall_2*, *Rise_2*}, representing seasonal cycles over two years, three asymmetries in amplitude and period length would be calculated for the sequence of segments, as (*i*) Fall_1 - Rise_1, (*ii*) Fall_2 - Rise_1, and (*iii*) Fall_2 - Rise_2. The asymmetries are referenced to Fall segments such that, for example, negative asymmetries mean that the amplitude (or period length) is greater in the Rise segment. The reason we calculated asymmetries between segments immediately before and

after the Fall segments is because we assumed that there is some degree of autocorrelation in the relational values that is both real and could provide useful information, but the underlying causal mechanisms are speculative at this point.

### 2.8 Application of approach

We applied the approach to evaluate the effect of LUC on XCO$_2$ by using the segment characteristics setting the

'S2' scenario as the reference time series and then following procedures outlined in Sect. 2.6 to match corresponding Rise and Fall segments in the S3 and S2 simulations. We then calculated the difference in the amplitude, period, and phase between matching segments, hereafter defined as the 'LUCeffect'. To evaluate the relative influence of the LUCeffect on changes in amplitude, period and phase, we transformed the LUCeffect to percentages by (a) dividing



the LUCeffect in amplitude by region-specific average amplitudes, and (b) dividing the LUCeffect in the period
length and phase (DOYstart, DOYend) by the region-specific average period lengths. We then pooled the absolute
values of the standardized LUCeffects for all regions, by model; the absolute values of LUCeffect was used because
we were more interested in any significant change, rather than a directional change in the metric values. We
conducted an Analysis of Variance to test for significant differences among models and type of LUCeffect
(amplitude, period, and phase), in terms of the percent LUCeffect, also setting significant differences at p=0.05. In
this manner, we were able to determine the relative importance of LUCeffect by metric and compare amongst
models.

### 3. Results

### 3.1 Satellite coverage and XCO$_2$ seasonal cycles

The satellite data coverage had sufficient temporal density to extract smooth seasonal cycles (Fig. 3), except during
Boreal Winter at high latitudes (> 50˚ N) and during the wet-season in Tropical Asia where there was clear evidence
of linear interpolation over large data gaps (Supplementary Material Fig. S2-S4). We had to exclude North America
Boreal and South America Tropical regions from all analyses because the data were too sparse and seasonal cycles
could not be derived. The mean number of satellite retrievals per day in 5˚ bins was greater than 1 when averaged
over a season, but the spatial distribution of the retrievals by month (Supplementary Material Fig. S2-S4) showed
that only portions of the TransCom regions were being represented with satellite observations. The lack of a
complete representative sample of satellite observations in a region suggests that the derived seasonal cycle will be
biased towards the XCO$_2$ observations in those sub-regions with greater coverage. We take this finding as a caveat,
but also demonstrate below that the derived seasonal cycles are a good representation of the general seasonal
dynamics in the data.
300       There were noticeable deviations (local minimums) from otherwise consistent Rise and Fall patterns during a
season (for example in North Africa in Fig. 3). We compared the seasonal cycles derived from DGVM XCO$_2$ co-
located with GOSAT retrievals against DGVM seasonal cycles using all simulated XCO$_2$ and complete coverage
(no-colocation). For the single DGVM studied in this side analysis, the local deviations were still evident in the
seasonal cycles that used data with complete coverage (Supplementary Material Fig. S5). We believe that these
deviations are not artefacts of the spatial distribution of satellite retrievals, but instead are true patterns in the XCO$_2$
seasonal cycle. However, the co-location sampling did appear to have a greater effect on the amplitudes and periods
in Southern Hemisphere regions, whereas the effect of co-location sampling was less influential in Northern
Hemisphere regions.
     The magnitude of the GOSAT seasonal cycle residual error, averaged over all regions, was 0.15 ± 1.02 ppm,
which was not a small fraction relative to the average amplitudes when taking into account the standard deviation.
However, the residuals, were normally and randomly distributed around zero (Supplementary Material Fig. S6),
which we took to suggest that there was no systematic bias and that the daily spatial variation in data coverage
averaged out, and what we derived was a realistic estimate of seasonal variation in XCO$_2$.





### 3.2 Latitudinal variation in XCO$_2$ seasonal cycle amplitudes

The amplitudes of XCO$_2$ seasonal cycles varied predictably with latitude (Fig. 4), and there were no significant differences in the latitudinal gradient between Rise and Fall segments for either GOSAT. Latitude explained between 82-84% of the variation in seasonal amplitudes in GOSAT, with the range taken from linear models of Rise and Fall segments (Fig. 4). There was an increase in amplitude of 0.74-0.77 ppm for every 10 degrees of latitude for GOSAT. Whereas the XCO$_2$ amplitudes exhibited a linear relationship with latitude, the in-situ flask samples of

CO$_2$ exhibited a log-linear relationship with latitude (Fig. 5; $R^2 = 0.90$, d.f.=45, F= 410.5, p < 0.001). Furthermore, the latitudinal gradient in seasonal amplitude for the CO$_2$ in-situ data was 1.25 ppm/10° latitude (Fig. 5), a ~65% increase compared to the amplitude gradient from GOSAT XCO$_2$. This results in a latitudinal range in seasonal amplitude of ~8 ppm for XCO$_2$ and ~19 ppm for surface CO$_2$. The dampened gradient in XCO$_2$ amplitude suggests substantial north-south atmospheric mixing, which is consistent with a previous study on the meridional versus

zonal contribution to XCO$_2$ via atmospheric transport (Keppel-Aleks et al., 2012). In addition, the in-situ sampling stations are located in such a way that they sample the 'background' atmosphere, which reduces the influence of local to regional terrestrial fluxes, and instead they provide seasonal cycles representative of hemispheric- and continental-scales. The contrast between the latitudinal gradient in amplitude between XCO$_2$ in this study and in-situ surface samples may therefore be even greater than reported here (Olsen and Randerson, 2004; Sweeney et al.,

330   2015).

Only LPX was able to simulate the GOSAT-derived latitudinal gradient (slope) in amplitude, but even in this model, the magnitudes of the amplitudes were consistently lower than GOSAT by ~1.5 ppm (Fig. 4). ORCHIDEE simulated the latitudinal gradient in amplitude reasonably well and CLM simulated a marginally stronger north-south gradient, whereas the gradient was much weaker in two models (OCN, VISIT) and there was no statistically

detectable amplitude gradient in LPJ. The evidently enhanced meridional mixing of total column CO$_2$ complicates an interpretation of the finding that most models simulated a weaker gradient in XCO$_2$ seasonal amplitude (Fig. 4). It makes it difficult to determine why models do not reproduce the latitudinal gradient in amplitude very well – for example, are the magnitudes of the fluxes in certain regions too low or too high, such that they offset the seasonal amplitudes in the region of interest after atmospheric transport? We offer suggestions in the Discussion that might

help answer these questions.

### 3.3 Latitudinal variation in XCO$_2$ seasonal cycle period

The period lengths of GOSAT XCO$_2$ seasonal cycles also varied predictably with latitude (Fig. 5) and there was no significant difference in the magnitude of the latitudinal gradients between Rise and Fall segments, although the direction of the gradient was positive for Rise segments and negative for Fall segments (Fig. 4). Latitude explained

between 67-73% of the latitudinal variation in period lengths in GOSAT seasonal cycles. From South to North, the period lengths of Rise segments increased by 10 days per 10° of latitude for GOSAT. From South to North, the period lengths of Fall segments had negative gradient and decreased by -9 days/10° latitude for GOSAT. The opposite gradient in period lengths of Rise and Fall segments implies that around 2° N, the asymmetry in period lengths reverse sign. North of this point of inversion in asymmetry, the period lengths of Rise segments are greater

than in Fall segments, with an increasing asymmetry as latitude increases. We suggest that the latitude of inversion of period asymmetry is a characteristic indicator of global atmospheric dynamics and biosphere productivity. As of yet, however, it is unclear if this point of inversion is relatively stable over time or if, instead, the point shifts in latitude among years or decades depending on the relative influence of source-sink dynamics in biospheres in the Northern and Southern Hemispheres.

Most models correctly simulated the satellite-derived latitudinal gradient in period, but LPJ and VISIT did not simulate statistically significant gradients in either Rise or Fall segments, and LPX could only reproduce the gradient for Rise segments, but not for Fall segments (Fig. 4). For CLM, OCN and ORCHIDEE, the simulated gradients were statistically similar to GOSAT and OCO-2, although the absolute period lengths differed by up to 25 days. The latitudinal gradient in period of $XCO_2$ seasonal cycles is emergent from the underlying timing and

duration of biosphere productivity, and as such, it serves as a high-level constraint on simulated dynamics. It may therefore be possible to add this emergent pattern as a benchmark to evaluate models that attempt to reproduce more direct indicators of biosphere activity, such as seasonal patterns in leaf area (Richardson et al., 2012), or primary production (Forkel et al., 2014).

### 3.4 GOSAT asymmetries in period and amplitude

The period asymmetry between Rise and Fall segments (Table 2) is clearer when comparing the periods of consecutive Rise and Fall segments (Fig. 6), taking the Fall segment as reference, as described in Sect. 2.7. The period asymmetries were in the same direction except for the Africa Northern, Africa Southern, and South America Temperate regions (Fig. 6A). The asymmetries exhibit stable patterns of consistent direction within many regions, and they also display quite a bit of interannual variation in the magnitude (or direction in some cases) of the

asymmetries themselves (Fig. 6A and 6B). For example, the standard deviation in period asymmetry averaged 11% of the region-averaged periods for GOSAT seasonal cycles, and it was greatest for the Africa Southern region (42%). For context, a 10% change amounts to a change in period asymmetry by 5-29 days, and as much as 73 days in the Africa Southern regions, which is certainly a remarkable change in the atmospheric signal. The period asymmetries can provide insight into the underlying terrestrial dynamics, for example, from interannual variation in

the duration of the carbon uptake period (Xia et al., 2015; Fu et al., 2017), but it is yet unclear how changes in carbon uptake period manifest to affect these patterns of asymmetry. Furthermore, one DGVM (ORCHIDEE) was able to simulate period asymmetries, consistent in direction, with that of the GOSAT record when using co-location sampling. Albeit, the magnitude of the period asymmetry for ORCHIDEE was about half that of GOSAT, but it does suggest that the surface fluxes from this DGVM were more realistic in timing and magnitude. All other models had

greater interannual variation in the direction of the asymmetry, with no other model reproducing the direction of asymmetry in all regions.

The amplitude asymmetries between consecutive Rise and Fall segments were more variable in the direction of the asymmetry for GOSAT (Fig. 6B). There was no consistent pattern in the direction or magnitude of the amplitude asymmetries within or among regions, but we did not investigate if there were annual patterns that were consistent

among regions. No model successfully reproduced the direction of asymmetry in amplitude across all regions in



all years. As of yet, the relevance of interannual variation in the asymmetries is speculative, but we do know that such variation is not simply due to data coverage (Supplementary Material Fig. S5), so there may be more insightful information in the signal.

### 3.5 Correlated biases between Rise and Fall segments

The correlations of model biases differed more among Northern and Southern Hemispheres (NH and SH, respectively) than among regions, so we present the following analyses not by region, but by NH and SH. The NH regions were comprised of Africa Northern, Europe, Eurasia Temperate, North America Temperate; the SH regions were comprised of Africa Southern, Australia, and South America Temperate. These analyses required data on both Rise and Fall segments, which eliminated the Asia Tropical and Eurasia Boreal regions from these analyses.

Among Rise and Fall segments, and among all models and regions, the model biases in amplitude were nearly perfectly correlated (NH $R^2 = 0.99$, d.f. = 28, t= 64.63, p < 0.001; SH $R^2 = 0.99$, d.f. = 16, t = 65.02, p < 0.001; Fig. 7a and 7e). Except for ORCHIDEE and CLM, which exhibited the smallest amplitude biases, the other models all had amplitudes that were too low. In the SH, there was a similar pattern of negative amplitude biases (Fig. 7e), with exception that CLM simulated amplitudes that were too large in two of three SH regions. The strong correlations

suggest that knowing the amplitude biases in one part of the seasonal cycle is sufficient to gain information about amplitudes in the missing part of the seasonal cycle. This might be particularly useful for constraining estimates of $XCO_2$ seasonal cycle patterns during timeframes that have poor satellite coverage (Boreal Winter, Tropical Wet Season). Furthermore, it is revealing that models which simulate amplitudes that are too low do so almost equally for both Rise and Fall segments, which is suggestive of a systematic bias in the sensitivity of the models to seasonal

changes in climate. Such systematic biases can be due to simulated fluxes that are overall lower in magnitude, or due to a pattern of spatio-temporal fluxes that end up offsetting or cancelling each other in the atmospheric domain, but we cannot yet definitively attribute the bias of individual models to one of these possible causes.

The average period biases of Rise and Fall segments were also strongly correlated, with a greater strength of correlation in the NH ($R^2 = 0.77$, d.f. = 22, t= -8.53, p < 0.001) than in the SH ($R^2 = 0.82$, d.f. = 21, t = -9.87, p <

0.001). In the NH, almost all models simulated periods that were too short in Rise segments and too long in Fall segments, in approximately equal and opposing amounts (Fig. 7b). In the SH, the period biases spanned both positive and negative values for both of the Fall and Rise segments, but also in approximately equal and opposing amounts of bias (Fig. 7f). There were only a few data points where the periods within a region were either biased (a) too short for Rise segments and also too short for Fall segments, or (b) where the Rise segment was biased too long

and the Fall segment also biased too long. These patterns are suggestive of underlying constraints that compensate for biases in periods, such that situation (a) and (b), from above, rarely occur. Such constraints are likely associated with the underlying drivers of the period of Rise and Fall segments. For instance, models that simulate growing seasons that are too long will likely simulate Fall-segment periods that are also too long, and as a consequence, the dormant season will be shortened, as will the periods of associated Rise segments. Within a given model, the

magnitude of compensating biases varied by region, so it is possible that biases in biosphere activity varied similarly by region. To incorporate such insights will require direct manipulation of the phenology represented by models, but



improving the emergent patterns in period to better match the satellite-derived $XCO_2$ seasonal cycles will bolster confidence in the model's ability to represent both fine-scale dynamics and the emergent large-scale atmospheric patterns.

**3.6 Application of Approach: LUCeffects on amplitude, period and phase metrics were non-trivial**

We describe the LUCeffect as the percent change in the Rise and Fall segment amplitude, period, and phase (DOYstart, DOYend) when LUC processes are included in model simulations, relative to seasonal cycle metrics when LUC was not included in simulations. Among all models and Rise and Fall segments, the average LUCeffect was largest on amplitude (mean 13.4%, or 0.37 ppm), but there were also non-trivial changes in the period (7.2%, or
13.2 days), and phase metrics of the DOYend (6.5%, or 11.4 days) and DOYstart (6.2%, or 11.4 days). An Analysis of Variance suggested that the LUCeffects did not significantly differ between Rise and Fall segments (F= 0.006, d.f.=1, p = 0.941), and that the specific model explained 16% of the variation (F= 15.183, d.f.=5, p < 0.001) and the metric explained only 5% of the variation (F= 7.8153, d.f.=3, p < 0.001). LPJ was an outlier in that it simulated larger LUCeffects in every metric (mean LUCeffect=18%), approximately 8% greater than other models. The
remaining variation in LUCeffect was explained by the larger LUCeffect on amplitude in LPX and VISIT (Fig. 8), whereas OCN simulated only marginally greater LUCeffects than CLM and ORCHIDEE. The LUCeffects were of similar magnitudes as the baseline interannual variation for these metrics, in terms of percent change, or greater in the case of the LUCeffect on amplitude (Table 3).

The importance of the LUCeffect on the amplitude of Rise and Fall segments was somewhat expected because
LUC directly affects the type of land cover simulated in the models, for example, by converting forest to pasture or pasture to forest and thereby influencing the magnitude of surface fluxes directly (Arneth et al., 2017). However, the effect of LUC on the temporal metrics of the seasonal cycle (period, phase) is typically understated in the literature. The LUCeffects on period and phase are of the same relative magnitude as is observed in two-decades of advancement in the start and end dates of the carbon uptake period based on atmospheric inversion studies (Fu et al.,
2017). It should not be a surprise then that LUC can affect the timing of surface fluxes, but this facet is often overlooked when the focus is solely on variability at annual or decadal timescales. At the very least, this work shows that land-surface modelers should consider the impact of LUC on the timing and duration of surface fluxes, in addition to its effect on the magnitude of the fluxes.

**4. Discussion**

**4.1 Utility of a segment analysis for analyzing cyclic time series**

We demonstrated that a segmentation analysis of satellite-derived $XCO_2$ seasonal cycles can generate direct estimates of amplitude, period, and phase at global and hemispheric scales, and that it can reveal stable patterns in the metrics which can be used as benchmarks to evaluate simulation models. There is obvious value in using standard statistics (RMSE, S.D., $R^2$, etc.) to characterize a time series and evaluate it against simulated
reproductions (e.g., 'Taylor diagrams'; Taylor, 2001; Supplementary Fig. S7). We do this too, but we argue that applying statistical measures of goodness-of-fit over the entire time series misses an opportunity to extract valuable



information from observational data and provide more direct measures of bias. Studies that have evaluated amplitude and period biases directly have been based on the mean harmonic of the seasonal cycle (Peng et al. 2015), which lacks interannual variation, and therefore does not fully represent the modeled biases. Furthermore, the

metrics for the asymmetric Rise and Fall patterns in seasonal cycles are not typically estimated, nor evaluated for bias. In the Europe region, for example, the internannual variation in amplitude (1.25 ppm) and period (25 days) is certainly not trivial (Supplementary Fig. S8), and if excluded in evaluations it would cause a biased assessment of what the models can and cannot do well, limiting the potential of such assessments to inform potential improvements.

465         Our study focused on the Rise and Fall segments in $XCO_2$ seasonal cycles, corresponding to periods when terrestrial ecosystems generally release and uptake carbon dioxide, respectively. Other studies might be more interested in shorter-term, pulse-type signals, such as the ability of models to simulate the effect of large scale fires or volcano eruptions in an atmospheric time series. In either case, the segmentation algorithm could help standardize and decompose model bias into its component parts of amplitude, period and phase biases.

**4.2 Asymmetries provide new insights into the interannual variation of atmospheric signals**

By definition, the asymmetries (Fig. 6) are not anomalies, but similarly, the amplitude asymmetries are directly related to underlying processes generating the imbalance in the amplitude and period between Rise and Fall segments. Most likely, the asymmetries reflect the difference in the magnitude or in the timing of fluxes during the growing season for Fall segments and phenological dormancy for Rise segments (Randerson et al., 1997). Whereas

the signature of the terrestrial biosphere may be a more dominant driver of the period asymmetries, the amplitude asymmetries may also be influenced by processes that the models simply do not simulate well, or in any sufficient manner in some cases, such as sub-seasonal representation of Fire and LUC (Earles et al., 2012) or volcano eruptions (Jones and Cox, 2001). The interannual variation in $XCO_2$ period and amplitude asymmetries are directly related the activity of terrestrial ecosystems, but questions remain – are the annual asymmetries in amplitudes or

periods evident of a global response to large-scale climate phenomena, such as the El Niño-Southern Oscillation? Do some regions dominate and influence the signal more than others? To what degree do the asymmetries in one region provide information about asymmetries in other regions, and can we infer dynamics in Boreal regions, for example, by analyzing atmospheric signals in regions where satellite coverage is more complete? The asymmetries offer a new level of information on atmospheric dynamics that is ripe for exploring.

**4.4 The effect of LUC on seasonal cycles is in addition to the effect on the long-term trend**

Much focus has been put on accurately characterizing component fluxes from land use and land cover change simulated by DGVMs (Pongratz et al., 2014; Calle et al., 2016), but we also show that LUC influences the atmospheric seasonal cycle period and phase at a level that is comparable to the reference rates of interannual variation in those metrics (Table 3). This underscores a complex problem of trying to simultaneously resolve the

contribution of LUC fluxes to the long-term trend in atmospheric $CO_2$ (Le Quéré et al., 2018), and also to represent realistic LUC effects on seasonal-scale biosphere activity (Betts et al., 2013; Bagley et al., 2014). For instance, when





land is converted from forest to pasture, the dominant land cover will affect the duration and timing of the surface

fluxes (Fleishcher et al., 2016) and this seems obvious on its own standing. However, DGVMs were not developed

during the era of satellite $XCO_2$ observations, and so the main issue of trying to resolve the effect of large-scale

changes in land use on both the long-term trend and seasonal cycle dynamics is not easily solved. But now that these

data are available, perhaps a new approach is necessary to take advantage of these large-scale benchmarks.

      The inclusion of LUC in the simulations, after including the contribution from fossil fuels and ocean,

resulted in a combined long-term trend estimate which was too large, by 0.07 to 1.72 ppm yr$^{-1}$, compared to the

long-term trend of GOSAT $XCO_2$ (2.16 ± 0.01 ppm yr$^{-1}$) (Supplementary Fig. S9). The GOSAT estimate is

comparable to an independent estimate of the long-term trend of $XCO_2$ from SCIAMACHY for the 2000s (1.95 ±

0.05 ppm yr$^{-1}$; Schneising et al., 2014). If we assume that this study's simulated long-term trends of fossil fuel

fluxes (4.44 ± 0.008 ppm yr$^{-1}$) and those of the ocean (-0.66 ± 0.0006 ppm yr$^{-1}$) are better constrained than the

trends from the land fluxes, then according to the GOSAT benchmark, the simulated land sink is too weak. Despite

the posibility that these simulated LUC fluxes are too high, the DGVM versions applied in this study do not simulate

a suite of land management processes (shifting cultivation, wood harvesting, pasture harvest, agriculture mgmt.) that

have been shown to increase the annual LUC flux by 20-60% (Arneth et al., 2017), further pointing to a simulated

land sink that is too weak. DGVM-based estimates of the terrestrial land sink have been compared against a residual

term in the global carbon budget that is taken as the average flux over a decade (Le Quéré et al., 2018), but perhaps

we are overlooking something here. The cumulative fluxes simulated by the models in this study (from 2002-2012)

resulted in a long-term trend that is at odds with the satellite record, and it is unclear why. We must therefore

attempt to reconcile biases in both the long-term trend and seasonal cycle dynamics if we are to use $XCO_2$, or other

integrated atmospheric measurements to constrain model dynamics, and not simply assess these patterns

independently.

### 4.5 Caveats, limitations and ways forward

The $XCO_2$ gradient in amplitude is approximately half the gradient in amplitude of in-situ surface $CO_2$. The

dampened $XCO_2$ gradient suggests the presence of strong meridional mixing, which complicates accurate attribution

of model biases to any specific bioregion. In effect, the $XCO_2$ seasonal cycle is comprised of the fluxes from all

regions to varying degrees (Olsen and Randerson, 2004; Sweeney et al., 2015; Lan et al., 2017). Given this,

simulating the atmospheric transport of the surface fluxes from all regions at once would allow us to both, (a) obtain

useable estimates of model bias and (b) to provide attribution to those biases. Indeed, the model biases were fully

described, but only in terms of $XCO_2$, not in terms of terrestrial surface fluxes themselves. An approach for

attribution of model bias in $XCO_2$ might be laid out similar to Liptak et al. (2017), wherein the surface fluxes from

each region (by year) undergo independent atmospheric transport. In a framework similar to this study, such

simulations might prove instrumental in determining the fractional contribution of each region's fluxes the $XCO_2$

seasonal cycle characteristics while also providing better guidance for model development.

      Model evaluations also showed that few models have low bias in all seasonal cycle metrics of amplitude,

period, and phasing of simulated $XCO_2$. An inherent requirement for reproducing the $XCO_2$ signal is that the land-



to-atmosphere fluxes are reasonable in magnitude, duration and timing *in all land regions*, or at the very least, in land regions with large vegetative areas that might disproportionately dominate the signal. Even though such

requirement may be necessary to simulate the amplitude asymmetries, this is an extreme level of proficiency that, simply, the models do not currently exhibit.

Lastly, the relative contribution of land, ocean and fossil fuel fluxes to the seasonal cycle differs by region, latitude, and time period (Randerson et al., 1997). This poses some concern because fossil fuel and cement fluxes are considered to have low uncertainty, but they may be biased too high in some regions (Saeki and Patra 2017),

affecting our interpretation of the contribution of simulated land fluxes to the seasonal cycle amplitudes, especially if the fossil fuel seasonal cycle signal is additive to (or offsets) the signal from the land fluxes. Other land uncertainties were not addressed in this study as it was not our intent to determine which DGVM had zero bias. Instead, we sought to extract unique patterns in the observed signals so that they may inform model development and subsequent evaluations in the future. Model improvements in their representation of important land processes

such as forest demography, wetland and permafrost dynamics, agriculture and land management, and a greater diversity of functional plant diversity are all on the horizon (Pugh et al., 2016; Fisher et al., 2018) and may further improve simulated atmospheric signals. The patterns in $XCO_2$ seasonal cycles are emergent from surface fluxes over the globe, and we foresee that a segment-based analysis of atmospheric seasonal cycles as a way to extract emergent patterns in the reference data to help guide future development and an improved understanding of the terrestrial

biosphere.

**Acknowledgements**

We thank Ehret and Zehe (2011) for their initial foray into alternative approaches to automate pattern matching in time series, which inspired this study. We thank the TRENDY Version 2 DGVM modelling community for their extensive efforts in continuing to advance model representation and making simulation data freely available. We

acknowledge the developers at NOAA ESRL that have maintained the C program of the ccgcrv algorithm and made it freely available. LC was supported by a National Aeronautics and Space Administration Earth and Space Science Fellowship (NASA ESSF 2016-2019). PKP acknowledges support from the Tougou (Theme B) project of the Ministry of Education, Culture, Sports, Science and Technology.

**Author Contribution**

LC, BP, and PKP conceived of the study. PKP conducted the atmospheric simulations. LC prepared and analyzed simulated and satellite data. LC developed the code for the segmentation algorithm. LC prepared the manuscript with contributions from all co-authors.

**Data Availability**

Data used in the analysis and code for the segmentation algorithm is freely-available from the GitHub repository

<https://github.com/lcalle/segmentTS>. The first version of the algorithm is archived and freely available at the Dryad Digital Repository <XXXX>.

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



**Table 1. Terrestrial ecosystem models from the TRENDY v.2 model inter-comparison used to simulate terrestrial Net Ecosystem Exchange.**

| Model | Abbrev. | Spatial Resolution | Land Surface Model | Fire Simulation | C-N coupled cycle | Source |
|---|---|---|---|---|---|---|
| Community Land Model v.4.5 | CLM | 2.5 X 2.5 | Yes | Yes | Yes | Lawrence et al. (2011) |
| Lund-Potsdam-Jena | LPJ | 0.5 X 0.5 | No | Yes | No | Sitch et al. (2003) |
| Land-surface Processes and exchanges | LPX | 1.0 X 1.0 | No | Yes | Yes | Prentice et al. (2011) |
| ORganizing Carbon and Hydrology in Dynamic EcosystEms | ORCHIDEE | 3.74 X 2.5 | Yes | Yes | No | Krinner et al. (2005) |
| ORCHIDEE with coupled C-N cycling | OCN | 1.0 X 1.0 | Yes | Yes | Yes | Zaehle and Friend (2010) |
| Vegetation Integrative SImulator for Trace gases | VISIT | 0.5 X 0.5 | No | Yes | Yes | Kato et al. (2013) |





**Table 2.** Signal characteristics for Rise and Fall segments of the GOSAT-derived XCO₂ seasonal cycles (2009-2012) by TransCom region. The timeframe of one Rise plus one Fall segment approximately equates to one year. North America Boreal and South America Tropical regions were excluded for lack of observations to derive signals for Rise or Fall segments.

| Region | Segment | Period (days) | | Amplitude (ppm) | |
|---|---|---|---|---|---|
| | | Fall | Rise | Fall | Rise |
| Africa Northern | 1,2 | 118 | 241 | 5.4 | 6.1 |
| | 3,4 | 130 | 229 | 5.5 | 5.2 |
| | 5,6 | 135 | 232 | 6.0 | 5.8 |
| | 7 | 135 | NA | 5.7 | NA |
| Africa Southern | 1,2 | 174 | 216 | 2.5 | 3.0 |
| | 3,4 | 131 | 131 | 4.0 | 3.6 |
| | 5,6 | 218 | 147 | 3.2 | 3.0 |
| Asia Tropical | 1,2 | NA | 194 | NA | 6.4 |
| | 3,4 | NA | 200 | NA | 7.5 |
| | 5,6 | NA | 190 | NA | 7.0 |
| Australia | 1,2 | 140 | 225 | 2.0 | 1.2 |
| | 3,4 | 136 | 209 | 2.0 | 2.5 |
| | 5,6 | 155 | 228 | 2.4 | 2.4 |
| Europe* | 2,1 | 115 | 236 | 6.8 | 8.0 |
| | 3,4 | 131 | 239 | 7.9 | 6.4 |
| | 5,6 | 132 | 244 | 6.1 | 7.4 |
| Eurasia Temperate* | 2,1 | 109 | 248 | 6.2 | 7.1 |
| | 3,4 | 108 | 255 | 7.2 | 6.4 |
| | 5,6 | 118 | 253 | 5.7 | 6.5 |
| Eurasia Boreal | 1,2 | 102 | NA | 10.9 | NA |
| | 3,4 | 100 | NA | 11.7 | NA |
| | 5,6 | 104 | NA | 11.2 | NA |
| North America Temperate | 1,2 | 129 | 235 | 6.4 | 6.8 |
| | 3,4 | 126 | 243 | 5.6 | 5.4 |
| | 5,6 | 127 | 233 | 6.0 | 5.3 |
| | 7 | 129 | NA | 5.6 | NA |
| South America Temperate | 1,2 | 232 | 91 | 2.1 | 2.0 |
| | 3,4 | 238 | 137 | 2.2 | 2.4 |
| | 5,6 | 234 | 154 | 2.9 | 2.6 |

* the first differentiable segment is a Rise segment, starting approximately ~100+ days
after the first segment in other regions because the initial drawdown (Fall segment) in the
region is a partial or incomplete segment.



**Table 3. The interannual variation (IAV) in XCO$_2$ seasonal cycle metrics, presented as the relative standard deviation (i.e., RSD or coefficient of variation) and the LUCeffect, defined as the change in the XCO$_2$ seasonal cycle metrics when land-use change is included in simulations, relative to simulations with only natural vegetation. The values for IAV and LUCeffect presented below are first calculated for each region and segment type (Rise, Fall), and then averaged over all regions, and models (for LUCeffect). The values for the phasing metrics (day of year, 'DOY') are calculated using the period as the divisor.**

| metric | GOSAT IAV (%) | LUCeffect (%) |
|---|---|---|
| amplitude | 12.3 | 14.2 |
| period | 14.5 | 7.5 |
| DOYstart | 9.3 | 6.5 |
| DOYend | 7.5 | 6.8 |





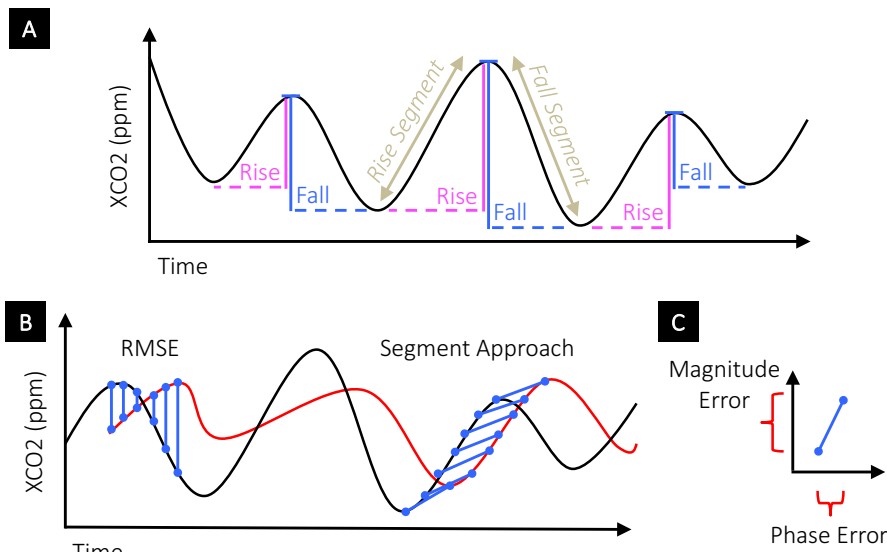

**Figure 1. Conceptual diagram for the segmentation analysis. (A) interannual variation in seasonal cycle amplitudes (vertical, solid colored lines) and periods (horizontal, dashed colored lines); such interannual variation may also differ among Rise and Fall segments. (B) a reference (black) and a modeled seasonal cycle (red) are compared using the Root Mean Squared Error (RMSE), which is taken as the difference in magnitude at the same exact time in reference and modeled seasonal cycles; in out-of-phase signals, the RMSE misrepresents bias; the segmentation approach matches segments in the reference and modeled seasonal cycles, Rise-to-Rise and Fall-to-Fall, so that the errors in magnitude and phase can be decomposed and directly represented (C).**





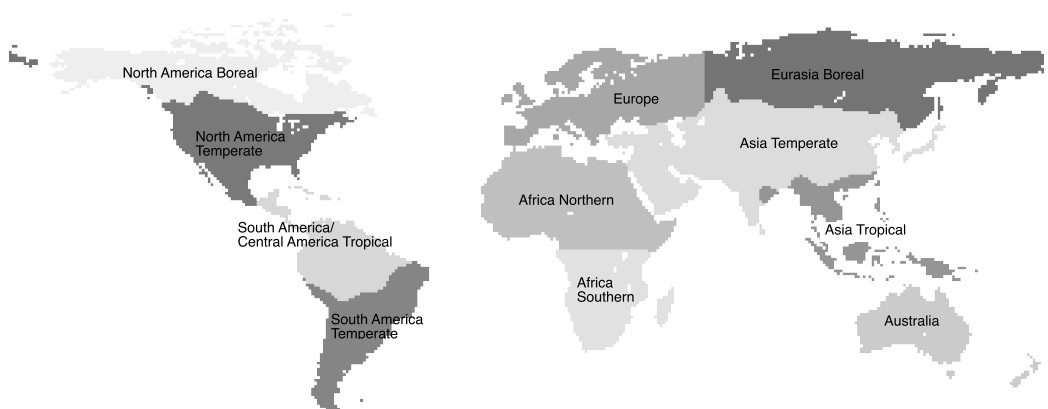

**Figure 2. TransCom region map.**




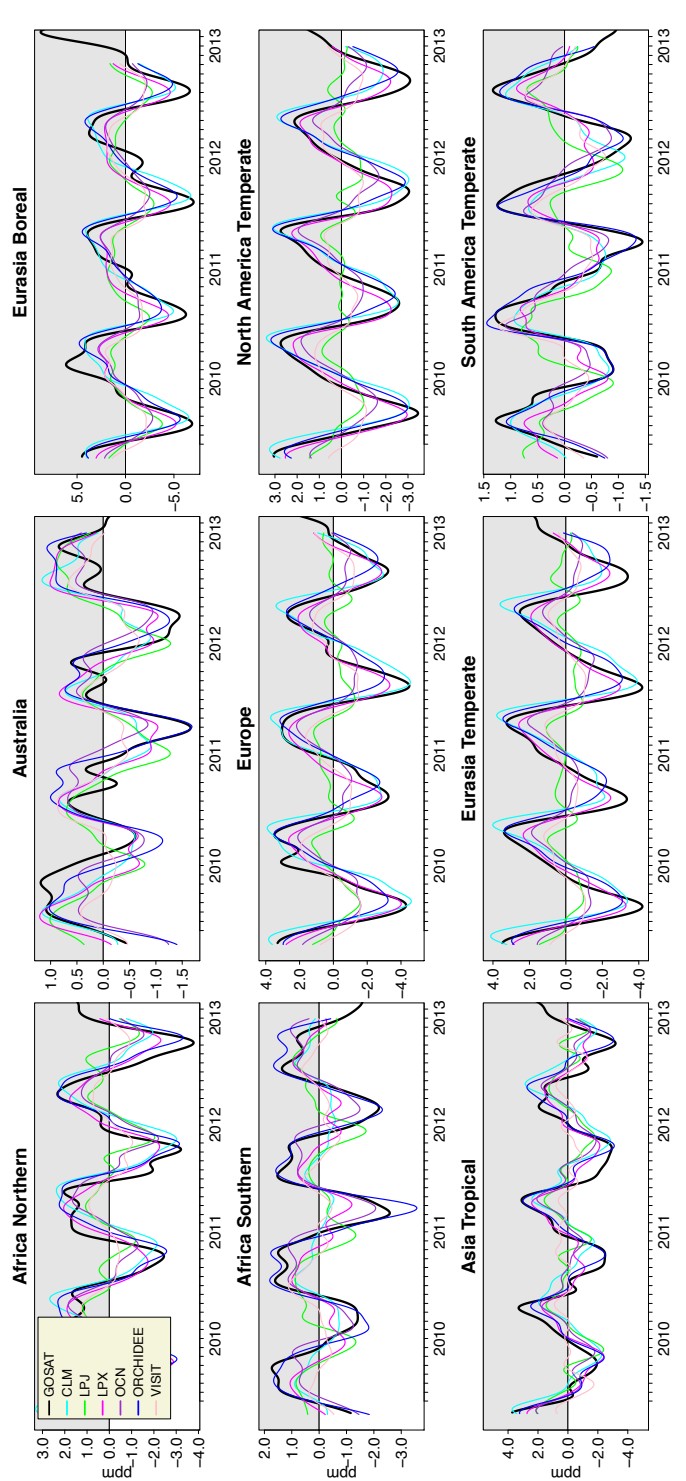

**Figure 3. Detrended XCO$_2$ seasonal cycles by TransCom region. Simulated seasonal cycles are the sum of transported fluxes from DGVM, Fossil Fuel and Ocean, but only the DGVM model name is listed.**



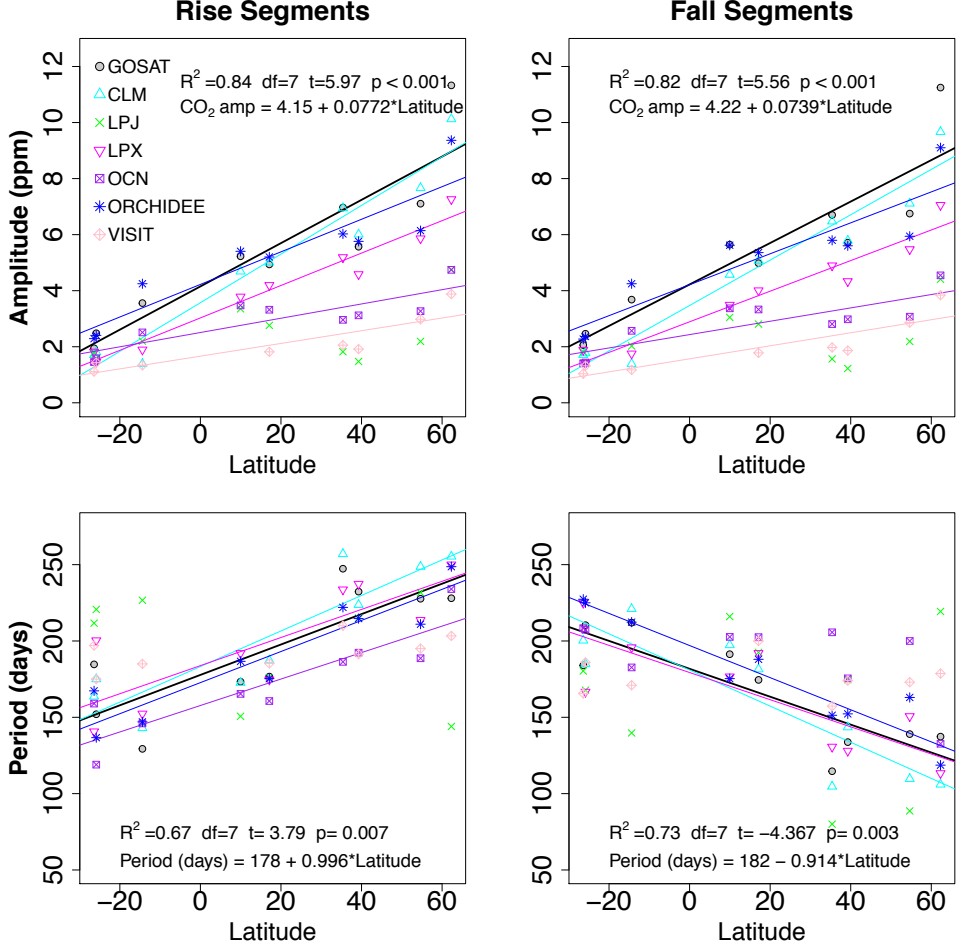

**Figure 4. Latitudinal variation in amplitude and period in Rise and Fall segments among TransCom regions, using the average latitude for each region. Linear regressions shown when significant (p < 0.05). Regression statistics and equation only given for GOSAT. OCO-2 data (orange, triangles) are from 2014-2018; all other data, including GOSAT, are from 2009-2012, corresponding to the date range of available simulation data.**


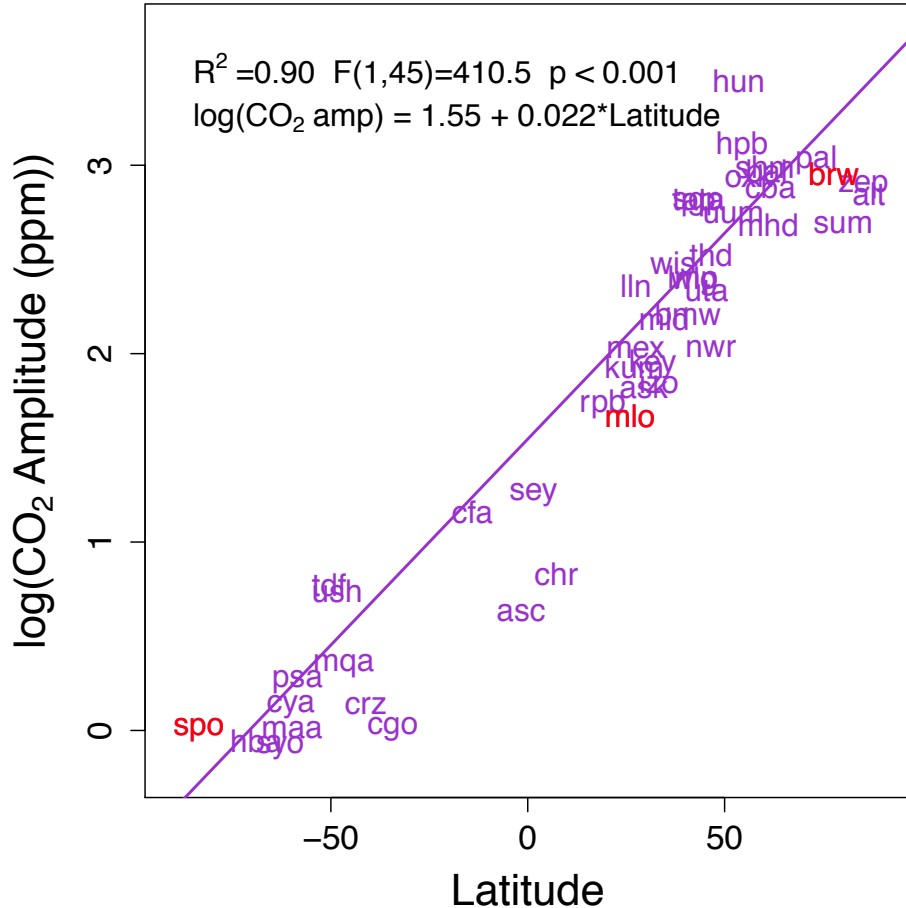

**Figure 5. Latitudinal variation in the amplitude for detrended in-situ surface CO₂ samples. Data are the average of peak-trough amplitudes for 2005-2015, only including sites with a minimum of 5 years of data. Points are labeled according to the three-letter code of the sampling station. South Pole (spo), Mauna Loa (mlo), and Barrow Island (brw) are highlighted in red for reference as these sites are commonly referenced in literature. The latitudinal range in surface site CO₂ seasonal amplitudes (~ 19 ppm), is more than 2 times the latitudinal range in seasonal amplitudes of XCO₂.**





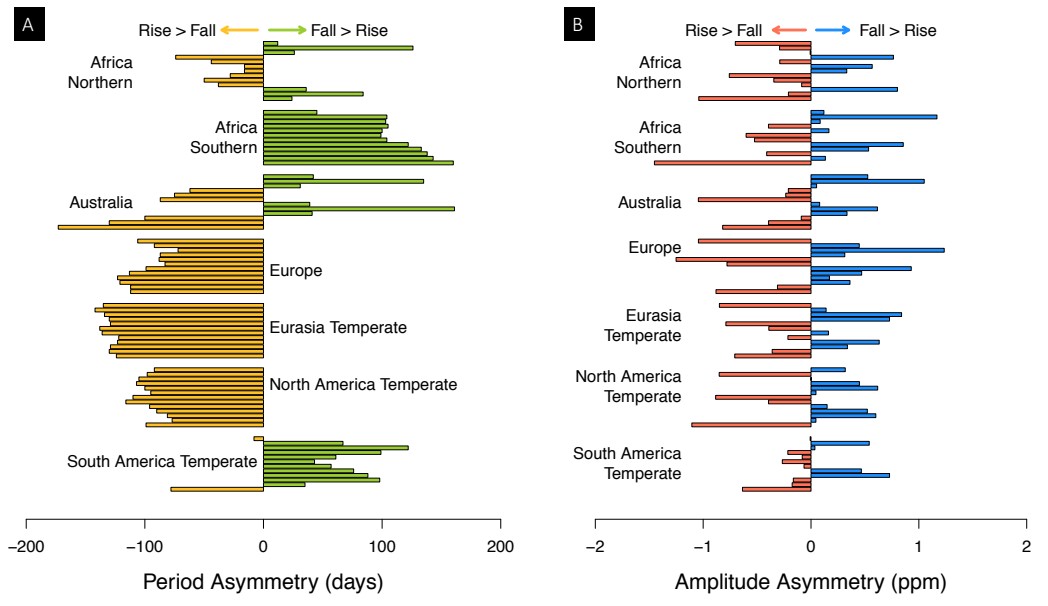

**Figure 6. Period asymmetries (A) and Amplitude asymmetries (B) in GOSAT XCO$_2$ seasonal cycles. Fall segments are taken as reference. Asymmetries are only shown for overlapping time periods.**





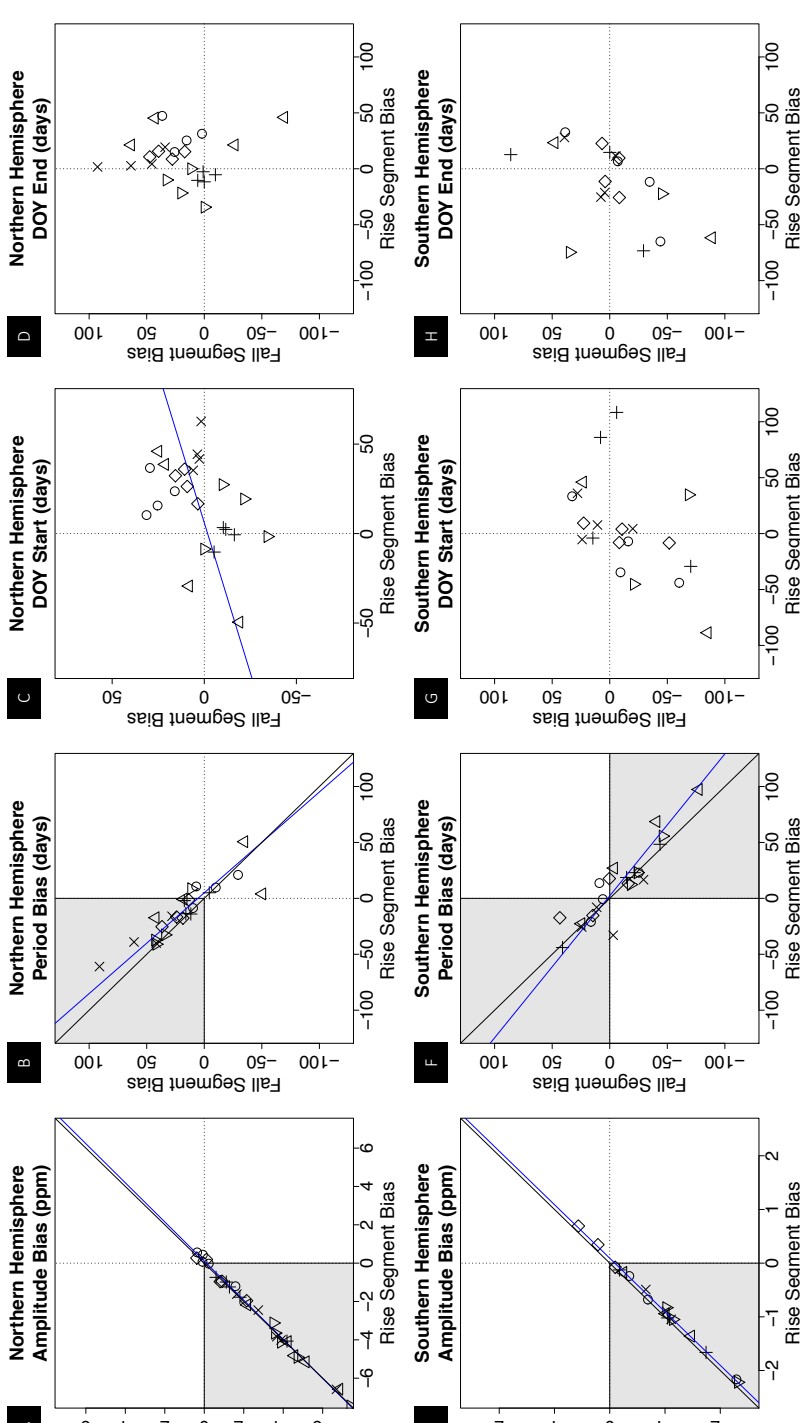

**Figure 7. Emergent correlations among biases for Rise (x-axes) and Fall (y-axes) segments model biases, using GOSAT XCO$_2$ as reference, for TransCom regions in the Northern Hemisphere (top row) and Southern Hemisphere (bottom row). Data points are the average bias by model (unique symbols, not shown) for a particular region. Data for the Eurasia Boreal and Asia Tropical regions were excluded for lack of data in both Rise and Fall segments. Diagonal black lines are the 1:1 correspondence lines, blue lines are significant linear correlations.**





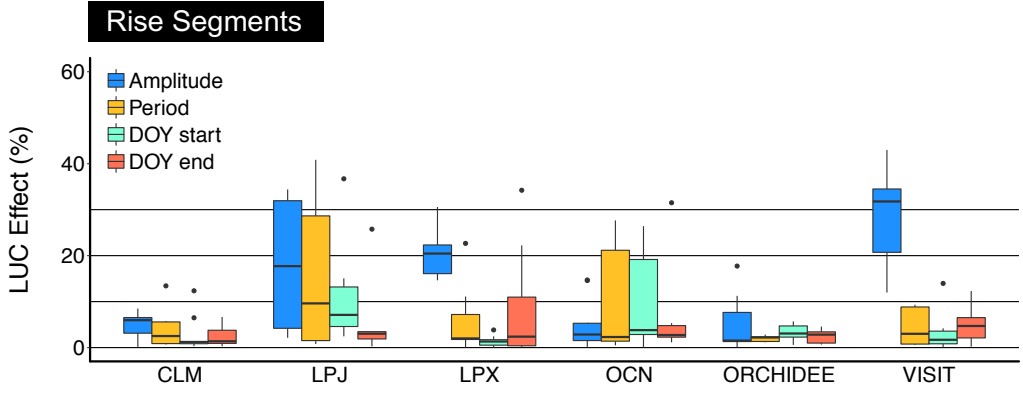

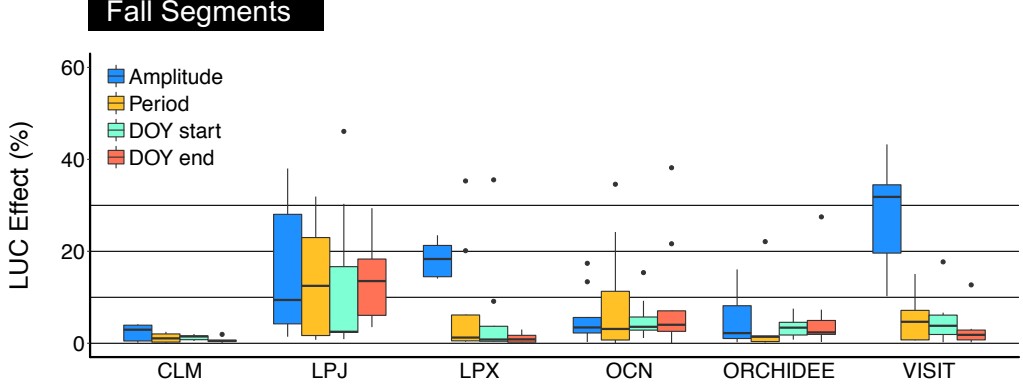

**Figure 8.** Land Use Change effect on amplitude, period, and day of year (DOY). The percentages were calculated from the difference in the metrics between simulations (S3-S2), scaled relative to amplitude and period of Rise and Fall segments for each region and model; DOY was scaled against the period.