# Peer review of "A segmentation algorithm for characterizing Rise and Fall segments in seasonal cycles: an application to XCO2 to estimate benchmarks and assess model bias"

_Atmospheric Measurement Techniques, 2018_

## Referee Comment (RC1) · Parazoo (Referee) · 2 Jan 2019

The study by Calle et al titled "A segmentation algorithm for characterizing Rise and Fall segments in seasonal cycles: an application to XCO2 to estimate benchmarks and assess model bias" examines unique aspects of the atmospheric co2 seasonal cycle using a new segmentation algorithm to characterize annual co2 rise and fall segments. A key application for carbon cycle science is attribution of model biases in co2 uptake patterns and underlying mechanisms by characterizing seasonal phase and amplitude biases in atmospheric co2 predictions. In particular, this directly addresses the prob-

lem of linking errors in the atmospheric co2 seasonal cycle amplitude change to corresponding errors in the timing and amplitude of biogenic co2 uptake. The method is well described, using conventional software and accepted smoothing techniques, and with enough detail that most interested users should be able to implement on their own (although some author assistance may be needed to account for local minima and maxima, which appears to be a science on its own). The applications are interesting and inciteful – I especially like the LUC sensitivity analysis. Some clarification is needed with respect to figures and in describing the main results, but the revisions are minor overall. I'm happy to recommend this paper for publication after addressing the comments below.

Minor comments

Sec 2.4 -> did you account for the averaging kernel in calculating xco2? I'm not sure how this varies geographically, but it may impose a different latitude gradient in xco2 than inferred from a simple pressure weighting

Sec 2.6.1 -> might be helpful to provide an example of time series with local minima/maxima and show how the algorithm differentiates these from seasonal mean values

Sec 2.7 -> I'm confused about the method to estimate the latitude gradient using the "average latitude of each TransCom region." Why not use the entire zonal average for each latitude band?

Sec 3.1, L290 -> the problem with using predefined transcom regions is the lack of coverage in critical sub-regions. I understand removing these regions from the analysis, but it seems archaic at this point to still use these regions. I will also point out that Eurasia Boreal has similar reduced coverage as NA Boreal (Fig S2), so it's odd that only the former region is analyzed

L316-318 -> this whole sentence is very confusing. Amplitude increases with latitude

in GOSAT. What else is there to say?

L318 -> Why is a range (0.74-0.77 ppm) for latitude slope reported? Convention is to report slope +/- uncertainty (e.g., 0.75 +/- 0.05 ppm). If referring to the upper and lower bounds due to errors in the slope, please specify.

Fig 4 -> The CO2amp and Period values in the figure don't make sense. Also why are there no points from 20-40N and between 40-50N? There is plenty of coverage according to Fig S2. I wonder if using zonal averaged to compute these latitude gradients would reduce this clumpiness

L321 -> It doesn't make sense to report a mean slope value (1.25 ppm / 10 deg lat) for a log-linear slope. Maybe just report the value for a certain latitude range (i.e., 30-40N)

L322-232 -> over what latitude range?

Sec 3.2 -> Why didn't you compare in situ observations to models?

L 348 -> Should mention somewhere that phases are equal at 2N. The switch in asymmetry in northern latitudes, specifically the rapid spring and slow fall transition at high northern latitudes, is consistent with findings in Parazoo et al (2016), who suggest that poleward transport of southern signals, which experience earlier spring and later fall, cause delayed but rapid spring drawdown and early but prolonged fall senescence in northern latitudes (Parazoo, N. C. et al, 2016, Detecting regional patterns of changing co2 flux in Alaska, PNAS)

L350-354 -> The question of whether the point of inversion changes over time due to biosphere activity could easily be answered with the model. Why not try this?

L357-359 and Fig 4 -> OCO-2 is mentioned in text and caption but I don't see any values plotted, and there is not mention of this data in the methods

Sec 3.4 -> A few comments: (1) The meaning of individual bars representing regional asymmetries in Fig 6 (10 bars total per region) is not explained in the text or the figure

caption. Assuming each bar represents 1 asymmetry between fall and rise segments, and there are 4 full years of GOSAT data, this should produce 7 bars. But 10 are shown, so maybe I'm misunderstanding the plot (thus the request for more cation detail). (2) There appears to be a mistake in identifying regions without all period asymmetries in the same direction: Africa Southern should be replaced with Australia? (3) All regions which have the same direction of period asymmetry are in the Northern Hemisphere, with the period of rise (fall transition) exceeding period of fall (spring transition), consistent with latitudinal results and 2N transition described earlier in the papers

L405: I do not understand how the statement "systematic bias in the sensitivity of models to seasonal changes in climate" explains model underestimates in amplitude of rise and fall segments. Please clarify with more detail and/or an example. It seems that the simplest explanation is that models underestimate growing season net uptake. Indeed, the situation in the next paragraph, in which models are too short in the rise (fall transition too short) and too long in the fall (spring transition too long) in the NH, is also consistent with a scenario in which models underestimate NH growing season net uptake (or perhaps, the timing of peak uptake is delayed)

Technical

L316 -> remove "either" at end of sentence?

Fig 3 -> need to move legend somewhere so it doesn't block time series

Figure 4 sub-panels need labels

L342 -> first sentence should refer to Fig 4

---

## Referee Comment (RC2) · Anonymous Referee #2 · 7 Jan 2019

Changes in the amplitude and phase of atmospheric carbon dioxide (CO2) reflect changes in geographical patterns and strengths of uptake and emissions and in atmospheric transport processes. Unfortunately, this information remains woefully underutilised in our quest to understand how uptake and emissions of CO2 are changing with time, even though a number of previous studies have made this point using a variety of statistical approaches . Here, the authors present a new methodology to characterise the rise and fall of seasonal cycles so that it can be used as a model metric. They apply this approach using column CO2 data from GOSAT. A focus on using

seasonal information to test carbon cycle models, particularly those models that are used in Earth system models, is very welcome.

Determining the amplitude and phase of a time series is a notoriously difficult problem, especially a time series with a superimposed time-dependent trend, normally requiring a lengthy time series to minimise the effect of edge effects. The GOSAT record runs from 2009 to present so I am curious while they curtailed their analysis at 2012. Armed with only a few seasonal cycles the authors will find it difficult to properly remove the lower frequency variations, which will arguably pervade the column measurements more so than surface measurements.

The authors have used a spectral method to remove short-term variations less than 80 days. It would be useful (for at this reader) to understand why they chose that value as a cut-off.

I thought that the math was presented in an unnecessarily complicated way. Surely, the second derivative and first derivative taken together are sufficient to determine the peak, trough and any saddle point found in the time series. Saddle points can be found in Arctic seasonal cycles, for instance.

Nevertheless, the method appears to be sound. The authors appear to focus on model evaluation instead of using the method to improving understanding of the carbon cycle. Consequently, there is little in the way of physical interpretation of the metrics in sections 3.2 and 3.3. How do the authors take into account the uncertainties associated with the column data?

For the model analysis, do the authors sample the model when/where there are observations?

Line 350: "We suggest that the latitude of the inversion of period asymmetry is a characteristic indicator of global atmospheric dynamics and biosphere productivity." It would be useful for the reader to understand the origin of this suggestion.

[Figure]

Line 360: "It may be possible to add this emergent pattern as a benchmark to evaluate models that attempt to reproduce more direct indicators of biosphere activity..." How important is atmospheric transport in determining zonal variations in this emergent pattern?

For the reasons outlined in the (balanced and frank) discussion I am left wondering how the metric will be used to "correct" models given the uncertainties associated with emissions from fossil fuel combustion and cement production. Could similar patterns emerge from nature and models for different reasons?

———————————————————

---

## Author Comment (AC1) · 5 Feb 2019

We thank both reviewers for their comments and suggestions. We know this paper is technical work and requires valuable time for such reviews. We think text added to the manuscript as a response to reviewers has made improvements and helped clarify assumptions and interpretation.

[Figure]

**1 Response to Reviewer 1**

Review1 comment1: Sec 2.4 -> did you account for the averaging kernel in calculating xco2? I'm not sure how this varies geographically, but it may impose a different latitude gradient in xco2 than inferred from a simple pressure weighting

Author's Response R1C1: We made the assumption that averaging kernel had a negligible effect on extracted seasonal cycles. No, we did not apply the GOSAT averaging kernel to our simulated $XCO_2$ calculations. It is true that the averaging kernel can affect point-to-point comparisons (Wunch et al. 2011). By most accounts, the difference in $XCO_2$ seasonal cycles forced by different DGVMs is quite large (magnitude and amplitude errors » 1 ppm, and phase errors on order of weeks). By comparison, the effect of an averaging kernel on extracted seasonal cycles is on the order of < 0.5 ppm (Lindqvist et al. 2015). Fig. 4; <doi:10.1186/s40562-017-0074-7>. Clarifying text was added as below to the methods section '2.1 Satellite $XCO_2$ data':

Author's changes to text R1C1: 'A note that satellite data have uncertainties of their own based on instrument noise, version of retrieval algorithm used to filter atmospheric effects, and averaging kernels (Yoshida et al. 2011, Lindqvist et al. 2015). We made the assumption that averaging kernel has a minimal effect on extracted seasonal cycles and we did not apply averaging kernels to the simulation data in this study. A full quantification of uncertainty in satellite-derived seasonal cycles is beyond the scope of this study, but such an analysis could be useful for benchmarking purposes as models continue to reduce large biases (» 1.0 ppm). Nevertheless, we make the assumption that lower biases are generally indicative of better model performance.'
Review1 comment2: Sec 2.6.1 -> might be helpful to provide an example of time se-
ries with local minima/maxima and show how the algorithm differentiates these from
seasonal mean values

Author's Response R1C2: We provide demonstrations of the algorithm per-
formance in the associated computer code for the algorithm, which is also
heavily annotated. We added clarifying text as below in the section '2.6
Technical Description..' to orient the reader to the additional resources.

Author's changes to text R1C2: 'We The computer code is annotated and
provides data used in this study with demonstrations for applying the algo-
rithm to remove local minima or maxima, and the categorization of seasonal
cycle segments.'

Review1 comment3: Sec 2.7 -> I'm confused about the method to estimate the latitude
gradient using the "average latitude of each TransCom region." Why not use the entire
zonal average for each latitude band?

Author's Response R1C3: The seasonal cycle metrics from the land re-
gions were sufficient to extract the relevant patterns for each latitude for ad-
dressing our main objectives. The main aim of our study is to evaluate the
quality of terrestrial biosphere model (LPJ DGVM) simulations. Although
the ocean fluxes also have seasonal variability but that can be considered
a minor contributor to the XCO2 seasonal cycle, relative to the flux seasonal
of land biosphere.

Review1 comment4: Sec 3.1, L290 -> the problem with using predefined transcom
regions is the lack of coverage in critical sub-regions. I understand removing these
regions from the analysis, but it seems archaic at this point to still use these regions. I

will also point out that Eurasia Boreal has similar reduced coverage as NA Boreal (Fig S2), so it's odd that only the former region is analyzed

> Author's Response R1C4: Yes, this is a good point. We do mention this is a caveat in the first paragraph of the discussion (section 3.1 'Satellite coverage...'). We had done a simple analysis using simulated XCO2 to assess the effect of data missing from sub-regions (Figure S5, in Supplement). In a few of the regions (i.e., Asia Tropical, Southern America Temperate) there were noticeable differences in seasonal cycles using co-location versus using all simulated data (no thinning). Analysis on smaller sub-regions would be useful, yes. We think this analysis is a good first step for comparing the DGVMs. So much of this type of analysis, and attribution of errors or fluxes to XCO2, is still related to the convolution fluxes in the near and far fields. Analysis on smaller regions does not help us much in identifying general patterns if we don't know the contributing field.

Review1 comment5: L316-318 -> this whole sentence is very confusing. Amplitude increases with latitude

> Author's Response R1C5: Agreed, we simplified the sentence as suggested.
>
> Author's changes to text R1C5: 'Seasonal amplitude varied predictably with latitude (Fig. 4).'

**2   Response to Reviewer 2**

Review2 comment1: Determining the amplitude and phase of a time series is a notoriously difficult problem, especially a time series with a superimposed time-dependent

trend, normally requiring a lengthy time series to minimise the effect of edge effects. The GOSAT record runs from 2009 to present so I am curious while they curtailed their analysis at 2012.

Author's Response R2C1: We clarified in the methods section '2.1 Satellite XCO2 data' that 'Satellite data for freely obtained only for 2009-2012 because it corresponded to the overlapping timeframe of available simulation data.'

Review2 comment2: Armed with only a few seasonal cycles the authors will find it difficult to properly remove the lower frequency variations, which will arguably pervade the column measurements more so than surface measurements. The authors have used a spectral method to remove short-term variations less than 80 days. It would be useful (for at this reader) to understand why they chose that value as a cut-off.

Author's Response R2C3: We used an 80-day cutoff value because it was specified as the standard value to remove short-term variations in seasonal cycle analyses when using the ccgcrv algorithm (Pickers and Manning 2015; also, described in <https://www.esrl.noaa.gov/gmd/ccgg/mbl/crvfit/crvfit.html>). To our understanding, and according to Thoning et al. (1989; pp 8558, 2nd para.; https://doi.org/10.1029/JD094iD06p08549), a low pass filter of 50 days was originally applied to remove shorter-frequency variations in the data that were unrelated to large-scale atmospheric mixing. That is, the intention of the low pass filter of 50-days was to retain month-scale variations in the atmospheric data. Apparently, the standard was since extended to 80-days for the short-term cutoff so that only variations that were evident, or maintained, for the time scale of 3-4 months were retained (3-4 month in the frequency domain is 4.56 cycles/yr). In the end, we thought such

a cutoff was suitable for this analysis because seasonal-scale variations are of general interest to terrestrial carbon cycle scientists. We added the following clarifying sentence to the text.  Author's changes to text R2C3: 'The cutoff for the short-term filter was set at the recommended value of 80 days (Thoning et al., 1989).  The short-term cutoff of 80-days retains data variations that are evident, or maintained, for the time scale of 3-4 months (4.56 cycles/yr).'

Review2 comment3: I thought that the math was presented in an unnecessarily complicated way. Surely, the second derivative and first derivative taken together are sufficient to determine the peak, trough and any saddle point found in the time series. Saddle points can be found in Arctic seasonal cycles, for instance.

Author's Response R2C3: Yes, we tend to agree. We had simplified the text description as such, but chose to also provide a mathematical description for those inclined towards symbols or for reproduction of the procedural steps of the algorithm without having to review the computer code. We would like to keep the mathematical level at this length, if there is no strong objection.

Review2 comment4: Nevertheless, the method appears to be sound.  The authors appear to focus on model evaluation instead of using the method to improving understanding of the carbon cycle. Consequently, there is little in the way of physical interpretation of the metrics in sections 3.2 and 3.3.

Author's Response R2C4: Yes, good point; we struggled with this ourselves given space limitations in describing the algorithm, the evaluation, and subsequent interpretation of models. We tried to outline future approaches in the Discussion for such interpretations.  The issue is that we deal with a
convolution of near- and far-field surface fluxes. We think the methods and algorithm presented in this study are a step forward towards the attribution of variation in the seasonal cycle metrics.

Review2 comment5: How do the authors take into account the uncertainties associated with the column data?

Author's Response R2C5: We use the Level-2 product that contains only high-quality and bias-adjusted data points. With regards to additional uncertainties in the satellite column data, we assume that uncertainties are random and normally distributed around zero, such that they average-out when taking the mean of all data points within a region. Spatially-averaged column uncertainties can be minor for seasonal cycle analyses if only considering the effect of the averaging kernel (0.15 ppm on average; Lindqvist et al. 2015 https://doi.org/10.5194/acp-15-13023-2015), but could amount to larger errors ( 1.5 ppm) if instrument noise, the main source of uncertainty, is also considered (Yoshida et al. 2011 doi:10.5194/amt‐4‐717‐2011). We added the following caveat to the text in the methods section: Author's changes to text R2C5: 'Satellite data have uncertainties of their own based on instrument noise, version of retrieval algorithm used to filter atmospheric effects, and averaging kernels (Yoshida et al. 2011, Lindqvist et al. 2015). A full quantification of uncertainty in satellite-derived seasonal cycles is beyond the scope of this study, but such an analysis could be useful for benchmarking purposes as models continue to reduce large biases (» 1.5 ppm). Nevertheless, we make the assumption that lower biases are generally indicative of better model performance.'

Review2 comment6: For the model analysis, do the authors sample the model when/where there are observations?

Author's Response R2C6: Yes, we use a co-location method to sample the simulated data. Clarifying text was updated as below, ref. Guerlet et al. 2013 https://doi.org/10.1002/jgrd.50332 Author's changes to text R2C6: 'We then used 'co-location' sampling of the ACTM XCO2 data to match the location and timeframe (13:00 hr local time) of observations, $\pm$ 5 days to account for (i.e., by averaging) sub-weekly transport errors (Guerlet et al., 2013).'

Review2 comment7: Line 350: "We suggest that the latitude of the inversion of period asymmetry is a characteristic indicator of global atmospheric dynamics and biosphere productivity." It would be useful for the reader to understand the origin of this suggestion.

Author's Response R2C7: We appreciate the suggestion. We replaced text and clarified as below: Author's changes to text R2C7: We hypothesize that the latitude at the point of inversion of period asymmetry is a characteristic indicator global atmospheric dynamics and biosphere productivity. Our rationale is that if (i) the primary driver of the period of drawdown (Fall) or release (Rise) in XCO2 seasonal cycles is the terrestrial biosphere, and (ii) DGVMs themselves simulate the terrestrial biosphere, then variation in the simulated point of inversion of asymmetry by different DGVMs suggests a strong influence of biosphere activity on this emergent pattern. The most obvious driver affecting the period being plant phenology. However, we already know that seasonal cycle in XCO2 is dominated by flux seasonality in land biosphere, with the ocean and fossil fuel emission seasonality plays only a secondary role.

Review2 comment8: Line 360: "It may be possible to add this emergent pattern as a benchmark to evaluate models that attempt to reproduce more direct indicators of

biosphere activity..." How important is atmospheric transport in determining zonal variations in this emergent pattern?

Author's Response R2C8: The effect of transport on zonal variation of this emergent pattern is likely to be large (Fig. 13 in Basu et al.; doi:10.1029/2011JD016124). Please note that the transport model (JAMSTEC's ACTM) used in this study generally performs well while evaluated against SF6 measurement, a tracer of atmospheric transport (https://www.atmos-chem-phys.net/11/12813/2011/; doi:10.1038/nature13721).

Review2 comment9: For the reasons outlined in the (balanced and frank) discussion I am left wondering how the metric will be used to "correct" models given the uncertainties associated with emissions from fossil fuel combustion and cement production. Could similar patterns emerge from nature and models for different reasons?

Author's Response R2C9: We know contribution of fossil fuel and cement (FFC) emissions will be less influential in seasonal cycle metrics. This is not to say that seasonality in FFC emissions is absent, but more so that the biosphere imprints a much larger signal on these patterns (Fig. 4; doi:10.1186/s40562-017-0074-7). Yes, similar patterns could emerge from nature and models for different reasons, and I think that the time-stepping of simulated processes in most models does not lend itself to realistic timeframes of surface fluxes that, ultimately, influence seasonal patterns in XCO2. For instance, the timing of fire and deforestation has a strong seasonality in the tropics (burning and clearing during dry seasons) and is implicit in the satellite data, but such seasonal dependence is lacking in model schemes. In this sense, the idea that these benchmarks will help correct models might be overstated. Perhaps it is better to suggest that models move toward these benchmark by first understanding the

limitations in direct comparisons of modeled surface fluxes to atmospheric XCO2. While potentially of great value to modelers, global ecosystem models were never designed with goal of using large scale emergent patterns in XCO2 as benchmarks so there are some basic hurdles to overcome.

---

## Author Response (AR2)

**1 Response to Reviewer 1**

We thank both reviewers for their comments and suggestions. We know this paper is technical work and requires valuable time for such reviews. We think text added to the manuscript as a response to reviewers has made improvements and helped clarify assumptions and interpretation.

Review1 comment1: Sec 2.4 -> did you account for the averaging kernel in calculating xco2? Im not sure how this varies geographically, but it may impose a different latitude gradient in xco2 than inferred from a simple pressure weighting

> Authors Response R1C1: We made the assumption that averaging kernel had a negligible effect on extracted seasonal cycles. No, we did not apply the GOSAT averaging kernel to our simulated XCO2 calculations. It is true that the averaging kernel can affect point-to-point comparisons (Wunch et al. 2011). By most accounts, the difference in XCO2 seasonal cycles forced by different DGVMs is quite large (magnitude and amplitude errors >> 1 ppm, and phase errors on order of weeks). By comparison, the effect of an averaging kernel on extracted seasonal cycles is on the order of ¡ 0.5 ppm (Lindqvist et al. 2015). Fig. 4; doi:10.1186/s40562-017-0074-7. Clarifying text was added as below to the methods section 2.1 Satellite XCO2 data:
> Authors changes to text R1C1: A note that satellite data have uncertainties of their own based on instrument noise, version of retrieval algorithm used to filter atmospheric effects, and averaging kernels (Yoshida et al. 2011, Lindqvist et al. 2015). We made the assumption that averaging kernel has a minimal effect on extracted seasonal cycles and we did not apply averaging kernels to the simulation data in this study. A full quantification of uncertainty in satellite-derived seasonal cycles is beyond the scope of this study, but such an analysis could be useful for benchmarking purposes as models continue to reduce large biases (>> 1.0 ppm). Nevertheless, we make the assumption that lower biases are generally indicative of better model performance.

Review1 comment2: Sec 2.6.1 -> might be helpful to provide an example of time series with local minima/maxima and show how the algorithm differentiates these from seasonal mean values

> Authors Response R1C2: We provide demonstrations of the algorithm performance in the associated computer code for the algorithm, which is also heavily annotated. We added clarifying text as below in the section 2.6 Technical Description.. to orient the reader to the additional resources.
> Authors changes to text R1C2: The computer code is annotated and provides data used in this study with demonstrations for applying the algorithm to remove local minima or maxima, and the categorization of seasonal cycle segments.

Review1 comment3: Sec 2.7 -> Im confused about the method to estimate the latitude gradient using the average latitude of each TransCom region. Why not use the entire zonal average for each latitude band?

> Authors Response R1C3: The seasonal cycle metrics from the land regions were sufficient to extract the relevant patterns for each latitude for addressing our main objectives. The main aim of our study is to evaluate the quality of terrestrial biosphere model (LPJ DGVM) simulations. Although the ocean fluxes also have seasonal variability but that can be considered a minor contributor to the XCO2 seasonal cycle, relative to the flux seasonal of land biosphere.

Review1 comment4: Sec 3.1, L290 -> the problem with using predefined transcom regions is the lack of coverage in critical sub-regions. I understand removing these regions from the analysis, but it seems archaic at this point to still use these regions. I will also point out that Eurasia Boreal has similar reduced coverage as NA Boreal (Fig S2), so its odd that only the former region is analyzed

> Authors Response R1C4: Yes, this is a good point. We do mention this is a caveat in the first paragraph of the discussion (section 3.1 Satellite coverage...). We had done a simple analysis using simulated XCO2 to assess the effect of data missing from sub-regions (Figure S5, in Supplement). In a few of the regions (i.e., Asia Tropical, Southern America Temperate) there were noticeable differences in seasonal cycles using co-location versus using all simulated data (no thinning). Analysis on smaller sub-regions would be useful, yes. We think this analysis is a good first step for comparing the DGVMs. So much of this type of analysis, and attribution of errors or fluxes to XCO2, is still related to the convolution fluxes in the near and far fields. Analysis on smaller regions does not help us much in identifying general patterns if we dont know the contributing field.

Review1 comment5: L316-318 -> this whole sentence is very confusing. Amplitude increases with latitude

Authors Response R1C5: Agreed, we simplified the sentence as suggested.

Authors changes to text R1C5: Seasonal amplitude varied predictably with latitude (Fig. 4).

Review1 comment6: L318 -> Why is a range (0.74-0.77 ppm) for latitude slope reported? Convention is to report slope +/- uncertainty (e.g., 0.75 +/- 0.05 ppm). If referring to the upper and lower bounds due to errors in the slope, please specify.

Authors Response R1C6: We understand the confusion. We updated the text to include slope estimate with uncertainty as mean +/- SE.

Authors changes to text R1C6: There was an increase in amplitude of 0.74 +/- 0.13 ppm ppm (mean +/- S.E.) for Rise Segments and 0.77 +/- 0.13 ppm for Fall Segments for every 10 degrees of latitude for GOSAT.

Review1 comment7: Fig 4 -> The CO2amp and Period values in the figure dont make sense. Also why are there no points from 20-40N and between 40-50N? There is plenty of coverage according to Fig S2. I wonder if using zonal averaged to compute these latitude gradients would reduce this clumpiness

Authors Response R1C7: We are not clear on why the CO2amp and Period values in Figure 4 do not make sense, could you clarify? The lack of data points in the latitudinal bands you suggest are because we used seasonal cycles from the transcom regions, taking the average latitude of the transcom region as the latitude for the analysis, plotted on the x-axes of Figure 4. This is the reason for the groupings of the data points.

Review1 comment8: L321 -> It doesnt make sense to report a mean slope value (1.25 ppm / 10 deg lat) for a log-linear slope. Maybe just report the value for a certain latitude range (i.e., 30-40N)

Authors Response R1C8: Good point, this was overlooked. We updated accordingly and reported that the log-linear slope indicates larger gradients at higher than at lower latitudes. We then also reported the latitudinal range in amplitudes between Equator and 70N for comparison between surface samples and XCO2.

Authors changes to text R1C8: Whereas the XCO2 amplitudes exhibited a linear relationship with latitude, in-situ flask samples of CO2 exhibited a log-linear relationship with latitude (Fig. 5; R2 = 0.90, d.f.=45, p < 0.001), which indicates larger amplitude gradients at higher than at lower latitudes. The difference results in a latitudinal range (Equator to 70N) in seasonal amplitude of ⌣ 7 ppm for XCO2 (taken as 70 * [0.077 + 1.95*0.013], as the largest possible amplitude gradient in XCO2; u + 1.95*S.E.) and ⌣ 17 ppm for surface CO2. The dampened gradient in XCO2 amplitude suggests substantial north-south atmospheric mixing...

Review1 comment9: L322-232 -> over what latitude range?

Authors Response R1C9: We specified that the latitudinal range in seasonal amplitude was compared between amplitude at the Equator and 70N for surface in-situ CO2 and XCO2 data. Text amended as in previous response R1C8.

Review1 comment10: Sec 3.2 -> Why didnt you compare in situ observations to models?

Authors Response R1C10: Although in-situ comparisons also have their value (Peng et al. 2015 in GBC), we were more interested in model abilities to reproduce larger scale patterns than at point-level

Review1 comment11: L 348 -> Should mention somewhere that phases are equal at 2N. The switch in asymmetry in northern latitudes, specifically the rapid spring and slow fall transition at high northern latitudes, is consistent with findings in Parazoo et al (2016), who suggest that poleward transport of southern signals, which experience earlier spring and later fall, cause delayed but rapid spring drawdown and early but prolonged fall senescence in northern latitudes (Parazoo, N. C. et al, 2016, Detecting regional patterns of changing co2 flux in Alaska, PNAS)

Authors Response R1C11: This is an interesting point. Yes, we mention the specific latitude at which the period in Rise and Fall segments are equal; we now clarify the text in this manner as below. We appreciate the reference as we had not seen this before. We think we understand what the reviewer is referring to S3: Long-range Transport analysis in Parazoo et al. (2016). Yes, we think that the latitude of equal phases (2N) should simply be thought of as an indicator of global activity. If the equal-phase latitude shifts north or south then one can infer that north-south mixing (i.e., poleward transport of signals) has changed substantially, and that, for example, northern seasonal cycle anomalies (in phase or amplitude) might have greater (or less) contribution from signals in the south. The procedure in Parazoo et al. (2016), which suggests zeroing fluxes by region to identify relative contribution of surface fluxes to regional signals, would be a good test to identify driving causes for potential shifts in the latitude of asymmetry inversion. We are also doing similar experiments to some extent, but think such analysis deserves its own focus. See changes added to the main text below.

Authors changes to text R1C11: The opposite gradient in period lengths of Rise and Fall segments implies that around 2N, the period of Rise and Fall segments are of equal duration. North of this point of inversion in asymmetry, the period lengths of Rise segments are greater than in Fall segments, with an increasing asymmetry as latitude increases... Nevertheless, a north or south shift in the latitude of inversion (i.e., 2N) would indicate that long-range transport of atmospheric signals, such as the poleward transport of southerly signals (Parazoo et al. 2016), has changed substantially. In which case, the relative contribution of long-range signals from southerly locations to seasonal cycle anomalies (in phase or amplitude) in northerly locations might be greater or less than expected.

Review1 comment12: L350-354 -> The question of whether the point of inversion changes over time due to biosphere activity could easily be answered with the model. Why not try this?

Authors Response R1C12: Yes, Reviewer2 also asked us to clarify and we updated the text to make explicit our hypotheses. See Authors response to Reviewer 2 comment R2C7. We inferred that biosphere activity affects the point of inversion because the DGVMs simulated different points of inversion. The DGVMs themselves simulate different spatio-temporal patterns of biosphere fluxes, so it seems to reason that the biosphere activity affects the point of inversion. As for extracting more information about this inversion point by conducting simulations with a single model – yes, this would be interesting. We are trying something along these lines, but we have yet to conclude that line research.

Review1 comment13: L357-359 and Fig 4 -> OCO-2 is mentioned in text and caption but I dont see any values plotted, and there is not mention of this data in the methods

Authors Response R1C13: This was an error. We did not include OCO-2 in the analyses for this paper. The text has been updated accordingly.

Review1 comment14: Sec 3.4 -> A few comments: (1) The meaning of individual bars representing regional asymmetries in Fig 6 (10 bars total per region) is not explained in the text or the figure.

Authors Response R1C14: The figure legend has been updated as below.

Authors changes to text R1C14: Figure 6. Period asymmetries (A) and Amplitude asymmetries (B) in GOSAT XCO2 seasonal cycles. The bars represent differences in amplitude or period for consecutive Fall (F) and Rise (R) segments, taking the Fall segment as reference. For example, if the time series follows the sequence F1, R1, F2, R2, F3, R3, (i.e., 3 seasonal cycles), then the difference between the first segment (F1) and the second segment (R2) is calculated as 'F1-R2'; a zero value would indicate that the metric (amplitude or period) were equal, whereas an asymmetry would be indicated by a positive (F1 ¿ R2) or negative (F1 ¡ R2) value. By definition, consecutive segments cannot be categorized as F-F or R-R. Asymmetry statistics are not traditional summaries, but nevertheless, they are characteristic and in some regions persistent patterns of the seasonal cycle that are undoubtedly influenced by biosphere activity.

We thank both reviewers for their comments and suggestions. We know this paper is technical work and requires valuable time for such reviews. We think text added to the manuscript as a response to reviewers has made improvements and helped clarify assumptions and interpretation.

**2 Response to Reviewer 2**

Review2 comment1: Determining the amplitude and phase of a time series is a notoriously difficult problem, especially a time series with a superimposed time-dependent trend, normally requiring a lengthy time series to minimise the effect of edge effects. The GOSAT record runs from 2009 to present so I am curious while they curtailed their analysis at 2012.

>Authors Response R2C1: We clarified in the methods section 2.1 Satellite XCO2 data that Satellite data was freely obtained and analyzed only for 2009-2012 because it corresponded to the overlapping timeframe of available simulation data.

Review2 comment2: Armed with only a few seasonal cycles the authors will find it difficult to properly remove the lower frequency variations, which will arguably pervade the column measurements more so than surface measurements. The authors have used a spectral method to remove short-term variations less than 80 days. It would be useful (for at this reader) to understand why they chose that value as a cut-off.

>Authors Response R2C3: We used an 80-day cutoff value because it was specified as the standard value to remove short-term variations in seasonal cycle analyses when using the ccgcrv algorithm (Pickers and Manning 2015; also, described in https://www.esrl.noaa.gov/gmd/ccgg/mbl/crvfit/crvfit.html). To our understanding, and according to Thoning et al. (1989; pp 8558, 2nd para.; https://doi.org/10.1029/JD094iD06p08549), a low pass filter of 50 days was originally applied to remove shorter-frequency variations in the data that were unrelated to large-scale atmospheric mixing. That is, the intention of the low pass filter of 50-days was to retain month-scale variations in the atmospheric data. Apparently, the standard was since extended to 80-days for the short-term cutoff so that only variations that were evident, or maintained, for the time scale of 3-4 months were retained (3-4 month in the frequency domain is 4.56 cycles/yr). In the end, we thought such a cutoff was suitable for this analysis because seasonal-scale variations are of general interest to terrestrial carbon cycle scientists. We added the following clarifying sentence to the text. Authors changes to text R2C3: The cutoff for the short-term filter was set at the recommended value of 80 days (Thoning et al., 1989). The short-term cutoff of 80-days retains data variations that are evident, or maintained, for the time scale of 3-4 months (4.56 cycles/yr).

Review2 comment3: I thought that the math was presented in an unnecessarily complicated way. Surely, the second derivative and first derivative taken together are sufficient to determine the peak, trough and any saddle point found in the time series. Saddle points can be found in Arctic seasonal cycles, for instance.

>Authors Response R2C3: Yes, we tend to agree. We had simplified the text description as such, but chose to also provide a mathematical description for those inclined towards symbols or for reproduction of the procedural steps of the algorithm without having to review the computer code. We would like to keep the mathematical level at this length, if there is no strong objection.

Review2 comment4: Nevertheless, the method appears to be sound. The authors appear to focus on model evaluation instead of using the method to improving understanding of the carbon cycle. Consequently, there is little in the way of physical interpretation of the metrics in sections 3.2 and 3.3.

>Authors Response R2C4: Yes, good point; we struggled with this ourselves given space limitations in describing the algorithm, the evaluation, and subsequent interpretation of models. We tried to outline future approaches in the Discussion for such interpretations. The issue is that we deal with a convolution of near- and far-field surface fluxes. We think the methods and algorithm presented in this study are a step forward towards the attribution of variation in the seasonal cycle metrics.

Review2 comment5: How do the authors take into account the uncertainties associated with the column data?

>Authors Response R2C5: We use the Level-2 product that contains only high-quality and bias-adjusted data points. With regards to additional uncertainties in the satellite column data, we assume that uncertainties are random and normally distributed around zero, such that they average-out when taking the mean of all data points within a region. Spatially-averaged column uncertainties can be minor for seasonal cycle analyses if only considering the effect of the averaging

kernel (0.15 ppm on average; Lindqvist et al. 2015 https://doi.org/10.5194/acp-15-13023-2015), but could amount to larger errors ( 1.5 ppm) if instrument noise, the main source of uncertainty, is also considered (Yoshida et al. 2011 https://doi.org/10.5194/amt-4-717-2011). We added the following caveat to the text in the methods section: Authors changes to text R2C5: Satellite data have uncertainties of their own based on instrument noise, version of retrieval algorithm used to filter atmospheric effects, and averaging kernels (Yoshida et al. 2011, Lindqvist et al. 2015). A full quantification of uncertainty in satellite-derived seasonal cycles is beyond the scope of this study, but such an analysis could be useful for benchmarking purposes as models continue to reduce large biases ($>>$ 1.5 ppm). Nevertheless, we make the assumption that lower biases are generally indicative of better model performance.

Review2 comment6: For the model analysis, do the authors sample the model when/where there are observations?

Authors Response R2C6: Yes, we use a co-location method to sample the simulated data. Clarifying text was updated as below, ref. Guerlet et al. 2013 https://doi.org/10.1002/jgrd.50332 Authors changes to text R2C6: We then used co-location sampling of the ACTM XCO2 data to match the location and timeframe (13:00 hr local time) of observations, 5 days to account for (i.e., by averaging) sub-weekly transport errors (Guerlet et al., 2013).

Review2 comment7: Line 350: We suggest that the latitude of the inversion of period asymmetry is a characteristic indicator of global atmospheric dynamics and biosphere productivity. It would be useful for the reader to understand the origin of this suggestion.

Authors Response R2C7: We appreciate the suggestion. We replaced text and clarified as below: Authors changes to text R2C7: We hypothesize that the latitude at the point of inversion of period asymmetry is a characteristic indicator global atmospheric dynamics and biosphere productivity. Our rationale is that if (i) the primary driver of the period of drawdown (Fall) or release (Rise) in XCO2 seasonal cycles is the terrestrial biosphere, and (ii) DGVMs themselves simulate the terrestrial biosphere, then variation in the simulated point of inversion of asymmetry by different DGVMs suggests a strong influence of biosphere activity on this emergent pattern. The most obvious driver affecting the period being plant phenology. However, we already know that seasonal cycle in XCO2 is dominated by flux seasonality in land biosphere, with the ocean and fossil fuel emission seasonality plays only a secondary role.

Review2 comment8: Line 360: It may be possible to add this emergent pattern as a benchmark to evaluate models that attempt to reproduce more direct indicators of biosphere activity... How important is atmospheric transport in determining zonal variations in this emergent pattern?

Authors Response R2C8: The effect of transport on zonal variation of this emergent pattern is likely to be large (Fig. 13 in Basu et al.; doi:10.1029/2011JD016124). Please note that the transport model (JAMSTECs ACTM) used in this study generally performs well while evaluated against SF6 measurement, a tracer of atmospheric transport (https://www.atmos-chem-phys.net/11/12813/2011/; doi:10.1038/nature13721).

Review2 comment9: For the reasons outlined in the (balanced and frank) discussion I am left wondering how the metric will be used to correct models given the uncertainties associated with emissions from fossil fuel combustion and cement production. Could similar patterns emerge from nature and models for different reasons?

Authors Response R2C9: We know contribution of fossil fuel and cement (FFC) emissions will be less influential in seasonal cycle metrics. This is not to say that seasonality in FFC emissions is absent, but more so that the biosphere imprints a much larger signal on these patterns (Fig. 4; doi:10.1186/s40562-017-0074-7). Yes, similar patterns could emerge from nature and models for different reasons, and we think that the time-stepping of simulated processes in most models does not lend itself to realistic timeframes of surface fluxes that, ultimately, influence seasonal patterns in XCO2. For instance, the timing of fire and deforestation has a strong seasonality in the tropics (burning and clearing during dry seasons) and is implicit in the satellite data, but such seasonal dependence is lacking in model schemes. In this sense, the idea that these benchmarks will help

correct models might be overstated. Perhaps it is better to suggest that models move toward these benchmark by first understanding the limitations in direct comparisons of modeled surface fluxes to atmospheric XCO2. While potentially of great value to modelers, global ecosystem models were never designed with goal of using large scale emergent patterns in XCO2 as benchmarks so there are some basic hurdles to overcome.

[revised manuscript text omitted]